# The non-convex Burer–Monteiro approach works on smooth semidefinite programs

**Nicolas Boumal**[⋆]
Department of Mathematics
Princeton University
nboumal@math.princeton.edu

**Vladislav Voroninski**[⋆]
Department of Mathematics
Massachusetts Institute of Technology
vvlad@math.mit.edu

**Afonso S. Bandeira**
Department of Mathematics and Center for Data Science
Courant Institute of Mathematical Sciences, New York University
bandeira@cims.nyu.edu

## Abstract

Semidefinite programs (SDPs) can be solved in polynomial time by interior point methods, but scalability can be an issue. To address this shortcoming, over a decade ago, Burer and Monteiro proposed to solve SDPs with few equality constraints via rank-restricted, non-convex surrogates. Remarkably, for some applications, local optimization methods seem to converge to global optima of these non-convex surrogates reliably. Although some theory supports this empirical success, a complete explanation of it remains an open question. In this paper, we consider a class of SDPs which includes applications such as max-cut, community detection in the stochastic block model, robust PCA, phase retrieval and synchronization of rotations. We show that the low-rank Burer–Monteiro formulation of SDPs in that class almost never has any spurious local optima.

This paper was corrected on April 9, 2018. Theorems 2 and 4 had the assumption that $\mathcal{M}$ (1) is a manifold. From this assumption it was stated that $\mathrm{T}_Y\mathcal{M} = \{\dot{Y} \in \mathbb{R}^{n \times p} : \mathcal{A}(\dot{Y}Y^\top + Y\dot{Y}^\top) = 0\}$, which is not true in general. To ensure this identity, the theorems now make the stronger assumption that gradients of the constraints $\mathcal{A}(YY^\top) = b$ are linearly independent for all $Y$ in $\mathcal{M}$. All examples treated in the paper satisfy this assumption. Appendix D gives details.

## 1 Introduction

We consider semidefinite programs (SDPs) of the form

$$f^* = \min_{X \in \mathbb{S}^{n \times n}} \langle C, X \rangle \quad \text{subject to} \quad \mathcal{A}(X) = b,\ X \succeq 0, \quad \text{(SDP)}$$

where $\langle C, X \rangle = \mathrm{Tr}(C^\top X)$, $C \in \mathbb{S}^{n \times n}$ is the symmetric cost matrix, $\mathcal{A} \colon \mathbb{S}^{n \times n} \to \mathbb{R}^m$ is a linear operator capturing $m$ equality constraints with right hand side $b \in \mathbb{R}^m$ and the variable $X$ is symmetric, positive semidefinite. Interior point methods solve (SDP) in polynomial time [Nesterov, 2004]. In practice however, for $n$ beyond a few thousands, such algorithms run out of memory (and time), prompting research for alternative solvers.

---

⋆The first two authors contributed equally.

If (SDP) has a compact search space, then it admits a global optimum of rank at most $r$, where $\frac{r(r+1)}{2} \leq m$ [Pataki, 1998, Barvinok, 1995]. Thus, if one restricts the search space of (SDP) to matrices of rank at most $p$ with $\frac{p(p+1)}{2} \geq m$, then the globally optimal value remains unchanged. This restriction is easily enforced by factorizing $X = YY^\top$ where $Y$ has size $n \times p$, yielding an equivalent quadratically constrained quadratic program:

$$q^* = \min_{Y \in \mathbb{R}^{n \times p}} \langle CY, Y \rangle \quad \text{subject to} \quad \mathcal{A}(YY^\top) = b. \tag{P}$$

In general, (P) is non-convex, making it a priori unclear how to solve it globally. Still, the benefits are that it is lower dimensional than (SDP) and has no conic constraint. This has motivated Burer and Monteiro [2003, 2005] to try and solve (P) using local optimization methods, with surprisingly good results. They developed theory in support of this observation (details below). About their results, Burer and Monteiro [2005, §3] write (mutatis mutandis):

> "*How large must we take $p$ so that the local minima of* (P) *are guaranteed to map to global minima of* (SDP)*? Our theorem asserts that we need only*[1] $\frac{p(p+1)}{2} > m$ *(with the important caveat that positive-dimensional faces of* (SDP) *which are 'flat' with respect to the objective function can harbor non-global local minima).*"

The caveat—the existence or non-existence of non-global local optima, or their potentially adverse effect for local optimization algorithms—was not further discussed.

In this paper, assuming $\frac{p(p+1)}{2} > m$, we show that if the search space of (SDP) is *compact* and if the search space of (P) is a *regularly defined smooth manifold*, then, for almost all cost matrices $C$, if $Y$ satisfies first- and second-order necessary optimality conditions for (P), then $Y$ is a global optimum of (P) and, since $\frac{p(p+1)}{2} \geq m$, $X = YY^\top$ is a global optimum of (SDP). In other words, first- and second-order necessary optimality conditions for (P) are also *sufficient* for global optimality—an unusual theoretical guarantee in non-convex optimization.

Notice that this is a statement about the optimization problem itself, not about specific algorithms. Interestingly, known algorithms for optimization on manifolds converge to *second-order critical points*,[2] regardless of initialization [Boumal et al., 2016].

For the specified class of SDPs, our result improves on those of [Burer and Monteiro, 2005] in two important ways. Firstly, for almost all $C$, we formally exclude the existence of spurious local optima.[3] Secondly, we only require the computation of second-order critical points of (P) rather than local optima (which is hard in general [Vavasis, 1991]). Below, we make a statement about computational complexity, and we illustrate the practical efficiency of the proposed methods through numerical experiments.

SDPs which satisfy the compactness and smoothness assumptions occur in a number of applications including Max-Cut, robust PCA, $\mathbb{Z}_2$-synchronization, community detection, cut-norm approximation, phase synchronization, phase retrieval, synchronization of rotations and the trust-region subproblem—see Section 4 for references.

**A simple example: the Max-Cut problem**

Given an undirected graph, Max-Cut is the NP-hard problem of clustering the $n$ nodes of this graph in two classes, $+1$ and $-1$, such that as many edges as possible join nodes of different signs. If $C$ is the adjacency matrix of the graph, Max-Cut is expressed as

$$\max_{x \in \mathbb{R}^n} \frac{1}{4} \sum_{i,j=1}^{n} C_{ij}(1 - x_i x_j) \quad \text{s.t.} \quad x_1^2 = \cdots = x_n^2 = 1. \tag{Max-Cut}$$

Introducing the positive semidefinite matrix $X = xx^\top$, both the cost and the constraints may be expressed linearly in terms of $X$. Ignoring that $X$ has rank 1 yields the well-known convex relaxation in the form of a semidefinite program (up to an affine transformation of the cost):

$$\min_{X \in \mathbb{S}^{n \times n}} \langle C, X \rangle \quad \text{s.t.} \quad \text{diag}(X) = \mathbf{1}, \, X \succeq 0. \quad \text{(Max-Cut SDP)}$$

If a solution $X$ of this SDP has rank 1, then $X = xx^\top$ for some $x$ which is then an optimal cut. In the general case of higher rank $X$, Goemans and Williamson [1995] exhibited the celebrated rounding scheme to produce approximately optimal cuts (within a ratio of .878) from $X$.

The corresponding Burer–Monteiro non-convex problem with rank bounded by $p$ is:

$$\min_{Y \in \mathbb{R}^{n \times p}} \langle CY, Y \rangle \quad \text{s.t.} \quad \text{diag}(YY^\top) = \mathbf{1}. \quad \text{(Max-Cut BM)}$$

The constraint $\text{diag}(YY^\top) = \mathbf{1}$ requires each row of $Y$ to have unit norm; that is: $Y$ is a point on the Cartesian product of $n$ unit spheres in $\mathbb{R}^p$, which is a smooth manifold. Furthermore, all $X$ feasible for the SDP have identical trace equal to $n$, so that the search space of the SDP is compact. Thus, our results stated below apply:

> *For $p = \lceil \sqrt{2n} \rceil$, for almost all $C$, even though* (Max-Cut BM) *is non-convex, any local optimum $Y$ is a global optimum (and so is $X = YY^\top$), and all saddle points have an escape (the Hessian has a negative eigenvalue).*

We note that, for $p > n/2$, the same holds for *all* $C$ [Boumal, 2015].

### Notation

$\mathbb{S}^{n \times n}$ is the set of real, symmetric matrices of size $n$. A symmetric matrix $X$ is positive semidefinite ($X \succeq 0$) if and only if $u^\top X u \geq 0$ for all $u \in \mathbb{R}^n$. For matrices $A, B$, the standard Euclidean inner product is $\langle A, B \rangle = \text{Tr}(A^\top B)$. The associated (Frobenius) norm is $\|A\| = \sqrt{\langle A, A \rangle}$. Id is the identity operator and $I_n$ is the identity matrix of size $n$.

## 2  Main results

Our main result establishes conditions under which first- and second-order necessary optimality conditions for (P) are sufficient for global optimality. Under those conditions, it is a fortiori true that global optima of (P) map to global optima of (SDP), so that local optimization methods on (P) can be used to solve the higher-dimensional, cone-constrained (SDP).

We now specify the necessary optimality conditions of (P). Under the assumptions of our main result below (Theorem 2), the search space

$$\mathcal{M} = \mathcal{M}_p = \{Y \in \mathbb{R}^{n \times p} : \mathcal{A}(YY^\top) = b\} \quad (1)$$

is a smooth and compact manifold of dimension $np - m$. As such, it can be linearized at each point $Y \in \mathcal{M}$ by a tangent space, differentiating the constraints [Absil et al., 2008, eq. (3.19)]:

$$\text{T}_Y \mathcal{M} = \{\dot{Y} \in \mathbb{R}^{n \times p} : \mathcal{A}(\dot{Y}Y^\top + Y\dot{Y}^\top) = 0\}. \quad (2)$$

Endowing the tangent spaces of $\mathcal{M}$ with the (restricted) Euclidean metric $\langle A, B \rangle = \text{Tr}(A^\top B)$ turns $\mathcal{M}$ into a Riemannian submanifold of $\mathbb{R}^{n \times p}$. In general, second-order optimality conditions can be intricate to handle [Ruszczyński, 2006]. Fortunately, here, the smoothness of both the search space (1) and the cost function

$$f(Y) = \langle CY, Y \rangle \quad (3)$$

make for straightforward conditions. In spirit, they coincide with the well-known conditions for unconstrained optimization. As further detailed in Appendix A, the Riemannian gradient $\text{grad} f(Y)$ is the orthogonal projection of the classical gradient of $f$ to the tangent space $\text{T}_Y \mathcal{M}$. The Riemannian Hessian of $f$ at $Y$ is a similarly restricted version of the classical Hessian of $f$ to the tangent space.

**Definition 1.** *A (first-order)* critical point *for* (P) *is a point* $Y \in \mathcal{M}$ *such that*

$$\operatorname{grad} f(Y) = 0, \qquad\qquad\qquad \text{(1st order nec. opt. cond.)}$$

*where* $\operatorname{grad} f(Y) \in \mathrm{T}_Y \mathcal{M}$ *is the Riemannian gradient at* $Y$ *of* $f$ *restricted to* $\mathcal{M}$. *A* second-order critical point *for* (P) *is a critical point* $Y$ *such that*

$$\operatorname{Hess} f(Y) \succeq 0, \qquad\qquad\qquad \text{(2nd order nec. opt. cond.)}$$

*where* $\operatorname{Hess} f(Y) \colon \mathrm{T}_Y \mathcal{M} \to \mathrm{T}_Y \mathcal{M}$ *is the Riemannian Hessian at* $Y$ *of* $f$ *restricted to* $\mathcal{M}$ *(a symmetric linear operator).*

**Proposition 1.** *All local (and global) optima of* (P) *are second-order critical points.*

*Proof.* See [Yang et al., 2014, Rem. 4.2 and Cor. 4.2]. $\qquad\qquad\qquad\qquad\qquad\qquad\square$

We can now state our main result. In the theorem statement below, "for almost all $C$" means potentially troublesome cost matrices form at most a (Lebesgue) zero-measure subset of $\mathbb{S}^{n \times n}$, in the same way that almost all square matrices are invertible. In particular, given any matrix $C \in \mathbb{S}^{n \times n}$, perturbing $C$ to $C + \sigma W$ where $W$ is a Wigner random matrix results in an acceptable cost matrix with probability 1, for arbitrarily small $\sigma > 0$.

**Theorem 2.** *Given constraints* $\mathcal{A} \colon \mathbb{S}^{n \times n} \to \mathbb{R}^m$, $b \in \mathbb{R}^m$ *and* $p$ *satisfying* $\frac{p(p+1)}{2} > m$, *if*

  *(i) the search space of* (SDP) *is compact; and*

  *(ii) the search space of* (P) *is a regularly-defined smooth manifold, in the sense that* $A_1 Y, \ldots, A_m Y$ *are linearly independent in* $\mathbb{R}^{n \times p}$ *for all* $Y \in \mathcal{M}$ *(see Appendix D),*

*then for almost all cost matrices* $C \in \mathbb{S}^{n \times n}$, *any second-order critical point of* (P) *is globally optimal. Under these conditions, if* $Y$ *is globally optimal for* (P), *then the matrix* $X = YY^\top$ *is globally optimal for* (SDP).

The assumptions are discussed in the next section. The proof—see Appendix A—follows directly from the combination of two intermediate results:

  1. If $Y$ is *rank deficient* and second-order critical for (P), then it is globally optimal and $X = YY^\top$ is optimal for (SDP); and

  2. If $\frac{p(p+1)}{2} > m$, then, for almost all $C$, every first-order critical $Y$ is rank-deficient.

The first step holds in a more general context, as previously established by Burer and Monteiro [2003, 2005]. The second step is new and crucial, as it allows to formally exclude the existence of spurious local optima, generically in $C$, thus resolving the caveat mentioned in the introduction.

The smooth structure of (P) naturally suggests using *Riemannian optimization* to solve it [Absil et al., 2008], which is something that was already proposed by Journée et al. [2010] in the same context. Importantly, known algorithms converge to second-order critical points regardless of initialization. We state here a recent computational result to that effect.

**Proposition 3.** *Under the numbered assumptions of Theorem 2, the Riemannian trust-region method (RTR) [Absil et al., 2007] initialized with any* $Y_0 \in \mathcal{M}$ *returns in* $\mathcal{O}(1/\varepsilon_g^2 \varepsilon_H + 1/\varepsilon_H^3)$ *iterations a point* $Y \in \mathcal{M}$ *such that*

$$f(Y) \leq f(Y_0), \qquad \|\operatorname{grad} f(Y)\| \leq \varepsilon_g, \qquad \text{and} \qquad \operatorname{Hess} f(Y) \succeq -\varepsilon_H \operatorname{Id}.$$

*Proof.* Apply the main results of [Boumal et al., 2016] using that $f$ has locally Lipschitz continuous gradient and Hessian in $\mathbb{R}^{n \times p}$ and $\mathcal{M}$ is a compact submanifold of $\mathbb{R}^{n \times p}$. $\qquad\square$

Essentially, each iteration of RTR requires evaluation of one cost and one gradient, a bounded number of Hessian-vector applications, and one projection from $\mathbb{R}^{n \times p}$ to $\mathcal{M}$. In many important cases, this projection amounts to Gram–Schmidt orthogonalization of small blocks of $Y$—see Section 4.

Proposition 3 bounds worst-case iteration counts for arbitrary initialization. In practice, a good initialization point may be available, making the local convergence rate of RTR more informative.

For RTR, one may expect superlinear or even quadratic local convergence rates near isolated local minimizers [Absil et al., 2007]. While minimizers are not isolated in our case [Journée et al., 2010], experiments show a characteristically superlinear local convergence rate in practice [Boumal, 2015]. This means high accuracy solutions can be achieved, as demonstrated in Appendix B.

Thus, under the conditions of Theorem 2, generically in $C$, RTR converges to global optima. In practice, the algorithm returns after a finite number of steps, and only *approximate* second-order criticality is guaranteed. Hence, it is interesting to bound the optimality gap in terms of the approximation quality. Unfortunately, we do not establish such a result for small $p$. Instead, we give an a posteriori computable optimality gap bound which holds for *all $p$* and for *all $C$*. In the following statement, the dependence of $\mathcal{M}$ on $p$ is explicit, as $\mathcal{M}_p$. The proof is in Appendix A.

**Theorem 4.** *Let $R < \infty$ be the maximal trace of any $X$ feasible for* (SDP). *For any $p$ such that $\mathcal{M}_p$ and $\mathcal{M}_{p+1}$ are smooth manifolds (even if $\frac{p(p+1)}{2} \leq m$) and for any $Y \in \mathcal{M}_p$, form $\tilde{Y} = [Y|0_{n\times 1}]$ in $\mathcal{M}_{p+1}$. The optimality gap at $Y$ is bounded as*

$$0 \leq 2(f(Y) - f^*) \leq \sqrt{R}\|\mathrm{grad} f(Y)\| - R\lambda_{\min}(\mathrm{Hess} f(\tilde{Y})). \tag{4}$$

*If all feasible $X$ have the same trace $R$ and there exists a positive definite feasible $X$, then the bound simplifies to*

$$0 \leq 2(f(Y) - f^*) \leq -R\lambda_{\min}(\mathrm{Hess} f(\tilde{Y})) \tag{5}$$

*so that $\|\mathrm{grad} f(Y)\|$ needs not be controlled explicitly. If $p > n$, the bounds hold with $\tilde{Y} = Y$.*

In particular, for $p = n + 1$, the bound can be controlled a priori: approximate second-order critical points are approximately optimal, for any $C$.[4]

**Corollary 5.** *Under the assumptions of Theorem 4, if $p = n + 1$ and $Y \in \mathcal{M}$ satisfies both $\|\mathrm{grad} f(Y)\| \leq \varepsilon_g$ and $\mathrm{Hess} f(Y) \succeq -\varepsilon_H \, \mathrm{Id}$, then $Y$ is approximately optimal in the sense that*

$$0 \leq 2(f(Y) - f^*) \leq \sqrt{R}\varepsilon_g + R\varepsilon_H.$$

*Under the same condition as in Theorem 4, the bound can be simplified to $R\varepsilon_H$.*

This works well with Proposition 3. For any $p$, equation (4) also implies the following:

$$\lambda_{\min}(\mathrm{Hess} f(\tilde{Y})) \leq -\frac{2(f(Y) - f^*) - \sqrt{R}\|\mathrm{grad} f(Y)\|}{R}.$$

That is, for any $p$ and any $C$, an approximate critical point $Y$ in $\mathcal{M}_p$ which is far from optimal maps to a comfortably-escapable approximate saddle point $\tilde{Y}$ in $\mathcal{M}_{p+1}$.

This suggests an algorithm as follows. For a starting value of $p$ such that $\mathcal{M}_p$ is a manifold, use RTR to compute an approximate second-order critical point $Y$. Then, form $\tilde{Y}$ in $\mathcal{M}_{p+1}$ and test the left-most eigenvalue of $\mathrm{Hess} f(\tilde{Y})$.[5] If it is close enough to zero, this provides a good bound on the optimality gap. If not, use an (approximate) eigenvector associated to $\lambda_{\min}(\mathrm{Hess} f(\tilde{Y}))$ to escape the approximate saddle point and apply RTR from that new point in $\mathcal{M}_{p+1}$; iterate. In the worst-case scenario, $p$ grows to $n + 1$, at which point all approximate second-order critical points are approximate optima. Theorem 2 suggests $p = \lceil \sqrt{2m} \rceil$ should suffice for $C$ bounded away from a zero-measure set. Such an algorithm already features with less theory in [Journée et al., 2010] and [Boumal, 2015]; in the latter, it is called the *Riemannian staircase*, for it lifts (P) floor by floor.

## Related work

Low-rank approaches to solve SDPs have featured in a number of recent research papers. We highlight just two which illustrate different classes of SDPs of interest.

Shah et al. [2016] tackle SDPs with linear cost and linear constraints (both equalities and inequalities) via low-rank factorizations, assuming the matrices appearing in the cost and constraints are

positive semidefinite. They propose a non-trivial initial guess to partially overcome non-convexity with great empirical results, but do not provide optimality guarantees.

Bhojanapalli et al. [2016a] on the other hand consider the minimization of a convex cost function over positive semidefinite matrices, without constraints. Such problems could be obtained from generic SDPs by penalizing the constraints in a Lagrangian way. Here too, non-convexity is partially overcome via non-trivial initialization, with global optimality guarantees under some conditions.

Also of interest are recent results about the harmlessness of non-convexity in low-rank matrix completion [Ge et al., 2016, Bhojanapalli et al., 2016b]. Similarly to the present work, the authors there show there is no need for special initialization despite non-convexity.

## 3   Discussion of the assumptions

Our main result, Theorem 2, comes with geometric assumptions on the search spaces of both (SDP) and (P) which we now discuss. Examples of SDPs which fit the assumptions of Theorem 2 are featured in the next section.

The assumption that the search space of (SDP),

$$\mathcal{C} = \{X \in \mathbb{S}^{n \times n} : \mathcal{A}(X) = b, X \succeq 0\}, \tag{6}$$

is *compact* works in pair with the assumption $\frac{p(p+1)}{2} > m$ as follows. For (P) to reveal the global optima of (SDP), it is necessary that (SDP) admits a solution of rank at most $p$. One way to ensure this is via the Pataki–Barvinok theorems [Pataki, 1998, Barvinok, 1995], which state that all *extreme points* of $\mathcal{C}$ have rank $r$ bounded as $\frac{r(r+1)}{2} \le m$. Extreme points are faces of dimension zero (such as vertices for a cube). When optimizing a linear cost function $\langle C, X \rangle$ over a compact convex set $\mathcal{C}$, at least one extreme point is a global optimum [Rockafellar, 1970, Cor. 32.3.2]—this is not true in general if $\mathcal{C}$ is not compact. Thus, under the assumptions of Theorem 2, there is a point $Y \in \mathcal{M}$ such that $X = YY^{\top}$ is an optimal extreme point of (SDP); then, of course, $Y$ itself is optimal for (P).

In general, the Pataki–Barvinok bound is tight, in that there exist extreme points of rank up to that upper-bound (rounded down)—see for example [Laurent and Poljak, 1996] for the Max-Cut SDP and [Boumal, 2015] for the Orthogonal-Cut SDP. Let $C$ (the cost matrix) be the negative of such an extreme point. Then, the unique optimum of (SDP) is that extreme point, showing that $\frac{p(p+1)}{2} \ge m$ is necessary for (SDP) and (P) to be equivalent for all $C$. We further require a strict inequality because our proof relies on properties of rank deficient $Y$'s in $\mathcal{M}$.

The assumption that $\mathcal{M}$ (eq. (1)) is a *regularly-defined smooth manifold* works in pair with the ambition that the result should hold for (almost) all cost matrices $C$. The starting point is that, for a given non-convex smooth optimization problem—even a quadratically constrained quadratic program—computing local optima is hard in general [Vavasis, 1991]. Thus, we wish to restrict our attention to efficiently computable points, such as points which satisfy first- and second-order KKT conditions for (P)—see [Burer and Monteiro, 2003, §2.2] and [Ruszczyński, 2006, §3]. This only makes sense if global optima satisfy the latter, that is, if KKT conditions are necessary for optimality. A global optimum $Y$ necessarily satisfies KKT conditions if *constraint qualifications* (CQs) hold at $Y$ [Ruszczyński, 2006]. The standard CQs for equality constrained programs are Robinson's conditions or metric regularity (they are here equivalent). They read as follows, assuming $\mathcal{A}(YY^{\top})_i = \langle A_i, YY^{\top} \rangle$ for some matrices $A_1, \ldots, A_m \in \mathbb{S}^{n \times n}$:

$$\text{CQs hold at } Y \text{ if } A_1 Y, \ldots, A_m Y \text{ are linearly independent in } \mathbb{R}^{n \times p}. \tag{7}$$

Considering almost all $C$, global optima could, a priori, be almost anywhere in $\mathcal{M}$. To simplify, we require CQs to hold at all $Y$'s in $\mathcal{M}$ rather than only at the (unknown) global optima. Under this condition, the constraints are independent at each point and ensure $\mathcal{M}$ is a smooth embedded submanifold of $\mathbb{R}^{n \times p}$ of codimension $m$ [Absil et al., 2008, Prop. 3.3.3]. Indeed, tangent vectors $\dot{Y} \in \mathrm{T}_Y \mathcal{M}$ (2) are exactly those vectors that satisfy $\langle A_i Y, \dot{Y} \rangle = 0$: under CQs, the $A_i Y$'s form a basis of the normal space to the manifold at $Y$.

Finally, we note that Theorem 2 only applies for *almost* all $C$, rather than all $C$. To justify this restriction, if indeed it is justified, one should exhibit a matrix $C$ that leads to suboptimal second-order critical points while other assumptions are satisfied. We do not have such an example. We do

observe that (Max-Cut SDP) on cycles of certain even lengths has a unique solution of rank 1, while the corresponding (Max-Cut BM) with $p = 2$ has suboptimal local optima (strictly, if we quotient out symmetries). This at least suggests it is not enough, for generic $C$, to set $p$ just larger than the rank of the solutions of the SDP. (For those same examples, at $p = 3$, we consistently observe convergence to global optima.)

## 4  Examples of smooth SDPs

The canonical examples of SDPs which satisfy the assumptions in Theorem 2 are those where the diagonal blocks of $X$ or their traces are fixed. We note that the algorithms and the theory continue to hold for complex matrices, where the set of Hermitian matrices of size $n$ is treated as a real vector space of dimension $n^2$ (instead of $\frac{n(n+1)}{2}$ in the real case) with inner product $\langle H_1, H_2 \rangle = \Re\{\mathrm{Tr}(H_1^* H_2)\}$, so that occurrences of $\frac{p(p+1)}{2}$ are replaced by $p^2$.

Certain concrete examples of SDPs include:

$$\min_X \langle C, X \rangle \text{ s.t. } \mathrm{Tr}(X) = 1, X \succeq 0; \qquad\qquad \text{(fixed trace)}$$

$$\min_X \langle C, X \rangle \text{ s.t. } \mathrm{diag}(X) = \mathbf{1}, X \succeq 0; \qquad\qquad \text{(fixed diagonal)}$$

$$\min_X \langle C, X \rangle \text{ s.t. } X_{ii} = I_d, X \succeq 0. \qquad\qquad \text{(fixed diagonal blocks)}$$

Their rank-constrained counterparts read as follows (matrix norms are Frobenius norms):

$$\min_{Y : n \times p} \langle CY, Y \rangle \text{ s.t. } \|Y\| = 1; \qquad\qquad \text{(sphere)}$$

$$\min_{Y : n \times p} \langle CY, Y \rangle \text{ s.t. } Y^\top = [y_1 \quad \cdots \quad y_n] \text{ and } \|y_i\| = 1 \text{ for all } i; \qquad \text{(product of spheres)}$$

$$\min_{Y : qd \times p} \langle CY, Y \rangle \text{ s.t. } Y^\top = [Y_1 \quad \cdots \quad Y_q] \text{ and } Y_i^\top Y_i = I_d \text{ for all } i. \qquad \text{(product of Stiefel)}$$

The first example has only one constraint: the SDP always admits an optimal rank 1 solution, corresponding to an eigenvector associated to the left-most eigenvalue of $C$. This generalizes to the trust-region subproblem as well.

For the second example, in the real case, $p = 1$ forces $y_i = \pm 1$, allowing to capture combinatorial problems such as Max-Cut [Goemans and Williamson, 1995], $\mathbb{Z}_2$-synchronization [Javanmard et al., 2016] and community detection in the stochastic block model [Abbe et al., 2016, Bandeira et al., 2016a]. The same SDP is central in a formulation of robust PCA [McCoy and Tropp, 2011] and is used to approximate the cut-norm of a matrix [Alon and Naor, 2006]. Theorem 2 states that for almost all $C$, $p = \lceil \sqrt{2n} \rceil$ is sufficient. In the complex case, $p = 1$ forces $|y_i| = 1$, allowing to capture problems where phases must be recovered; in particular, phase synchronization [Bandeira et al., 2017, Singer, 2011] and phase retrieval via Phase-Cut [Waldspurger et al., 2015]. For almost all $C$, it is then sufficient to set $p = \lfloor \sqrt{n} + 1 \rfloor$.

In the third example, $Y$ of size $n \times p$ is divided in $q$ slices of size $d \times p$, with $p \geq d$. Each slice has orthonormal rows. For $p = d$, the slices are orthogonal (or unitary) matrices, allowing to capture Orthogonal-Cut [Bandeira et al., 2016b] and the related problems of synchronization of rotations [Wang and Singer, 2013] and permutations. Synchronization of rotations is an important step in simultaneous localization and mapping, for example. Here, it is sufficient for almost all $C$ to let $p = \lceil \sqrt{d(d+1)q} \rceil$.

SDPs with constraints that are combinations of the above examples can also have the smoothness property; the right-hand sides 1 and $I_d$ can be replaced by any positive definite right-hand sides by a change of variables. Another simple rule to check is if the constraint matrices $A_1, \ldots, A_m \in \mathbb{S}^{n \times n}$ such that $\mathcal{A}(X)_i = \langle A_i, X \rangle$ satisfy $A_i A_j = 0$ for all $i \neq j$ (note that this is stronger than requiring $\langle A_i, A_j \rangle = 0$), see [Journée et al., 2010].

## 5  Conclusions

The Burer–Monteiro approach consists in replacing optimization of a linear function $\langle C, X \rangle$ over the convex set $\{X \succeq 0 : \mathcal{A}(X) = b\}$ with optimization of the quadratic function $\langle CY, Y \rangle$ over the

non-convex set $\{Y \in \mathbb{R}^{n \times p} : \mathcal{A}(YY^\top) = b\}$. It was previously known that, if the convex set is compact and $p$ satisfies $\frac{p(p+1)}{2} \geq m$ where $m$ is the number of constraints, then these two problems have the same global optimum. It was also known from [Burer and Monteiro, 2005] that spurious local optima $Y$, if they exist, must map to special faces of the compact convex set, but without statement as to the prevalence of such faces or the risk they pose for local optimization methods. In this paper we showed that, if the set of $X$'s is compact and the set of $Y$'s is a regularly-defined smooth manifold, and if $\frac{p(p+1)}{2} > m$, then for almost all $C$, the non-convexity of the problem in $Y$ is benign, in that all $Y$'s which satisfy second-order necessary optimality conditions are in fact globally optimal.

We further reference the Riemannian trust-region method [Absil et al., 2007] to solve the problem in $Y$, as it was recently guaranteed to converge from any starting point to a point which satisfies second-order optimality conditions, with global convergence rates [Boumal et al., 2016]. In addition, for $p = n + 1$, we guarantee that approximate satisfaction of second-order conditions implies approximate global optimality. We note that the $1/\varepsilon^3$ convergence rate in our results may be pessimistic. Indeed, the numerical experiments clearly show that high accuracy solutions can be computed fast using optimization on manifolds, at least for certain applications.

Addressing a broader class of SDPs, such as those with inequality constraints or equality constraints that may violate our smoothness assumptions, could perhaps be handled by penalizing those constraints in the objective in an augmented Lagrangian fashion. We also note that, algorithmically, the Riemannian trust-region method we use applies just as well to nonlinear costs in the SDP. We believe that extending the theory presented here to broader classes of problems is a good direction for future work.

## Acknowledgment

VV was partially supported by the Office of Naval Research. ASB was supported by NSF Grant DMS-1317308. Part of this work was done while ASB was with the Department of Mathematics at the Massachusetts Institute of Technology. We thank Wotao Yin and Michel Goemans for helpful discussions.

## Footnotes

[1]The condition on $p$ and $m$ is slightly, but inconsequentially, different in [Burer and Monteiro, 2005].

[2]Second-order critical points satisfy first- and second-order necessary optimality conditions.

[3]Before Prop. 2.3 in [Burer and Monteiro, 2005], the authors write: "The change of variables $X = YY^\top$ does not introduce any extraneous local minima." This is sometimes misunderstood to mean (P) does not have spurious local optima, when it actually means that the local optima of (P) are in exact correspondence with the local optima of "(SDP) *with the extra constraint* $\mathrm{rank}(X) \leq p$," which is also non-convex and thus also liable to having local optima. Unfortunately, this misinterpretation has led to some confusion in the literature.

[4]With $p = n + 1$, problem (P) is no longer lower dimensional than (SDP), but retains the advantage of not involving a positive semidefiniteness constraint.

[5]It may be more practical to test $\lambda_{\min}(S)$ (14) rather than $\lambda_{\min}(\mathrm{Hess} f)$. Lemma 7 relates the two. See [Journée et al., 2010, §3.3] to construct escape tangent vectors from $S$.

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
