[Supplementary Material]

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

[6] For non-symmetric $B \in \mathbb{R}^{n \times n}$, note that $\mathcal{A}(B) = \mathcal{A}\left(\frac{B + B^\top}{2}\right)$.

[7] For the Max-Cut SDP for example, $\mathcal{A} = \mathrm{diag}$ and $\mu = \mathrm{diag}(CYY^\top)$.

[8] Eq. (13) is equivalent to $G\mu = \mathcal{A}(CYY^\top)$, where $G_{ij} = \langle A_i Y, A_j Y \rangle$. For all $Y \in \mathcal{M}$, $G$ is invertible since $A_1 Y, \ldots, A_m Y$ are linearly independent. Hence, $\mu = G^{-1}\mathcal{A}(CYY^\top)$ is differentiable in $Y$ at $Y \in \mathcal{M}$.

[9]Downloaded from: http://web.stanford.edu/~yyye/yyye/Gset/ on June 6, 2016.

[10]Matlab R2015a on $2 \times 6$ cores processors with hyperthreading, Intel(R) Xeon(R) CPU E5-2640 @ 2.50GHz, 256Gb RAM, Springdale Linux 6.

[11]On Graph 77, running CVX leads to Matlab error "Number of elements exceeds maximum flint $2^{53} - 1$."

[12] If $\dot{Y} \in \mathrm{T}_Y \mathcal{M}$, by definition, there exists a smooth curve $\gamma : \mathbb{R} \to \mathcal{M}$ such that $\gamma(0) = Y$ and $\gamma'(0) = \dot{Y}$. Since $\gamma(t) \in \mathcal{M}$ for all $t$, we have $\mathcal{A}(\gamma(t)\gamma(t)^\top) = b$ for all $t$. Differentiating on both sides with respect to $t$ and evaluating at 0 gives $\mathcal{A}(\dot{Y}Y^\top + Y\dot{Y}^\top) = 0$.

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

# A  Proofs and additional lemmas

We start by working out explicit formulas for the Riemannian gradient and Hessian which appear in Definition 1. Let $\mathrm{Proj}_Y \colon \mathbb{R}^{n \times p} \to \mathrm{T}_Y \mathcal{M}$ be the orthogonal projector to the tangent space at $Y$ (eq. (2)), and let

$$\nabla f(Y) = 2CY, \qquad\qquad \nabla^2 f(Y)[\dot{Y}] = 2C\dot{Y} \qquad\qquad (8)$$

be the (Euclidean) gradient and Hessian of the cost function (3). The Riemannian gradient and Hessian of $f$ on $\mathcal{M}$ are related to these as follows [see Absil et al., 2008, eqs (3.37), (5.15)]:

$$\mathrm{grad} f(Y) = \mathrm{Proj}_Y \nabla f(Y), \qquad\qquad (9)$$

$$\forall \dot{Y} \in \mathrm{T}_Y \mathcal{M}, \quad \mathrm{Hess} f(Y)[\dot{Y}] = \mathrm{Proj}_Y \mathrm{D}\left(Y \mapsto \mathrm{grad} f(Y)\right)(Y)[\dot{Y}]. \qquad (10)$$

Let us focus on the gradient first. Since $\mathrm{grad} f(Y)$ is a tangent vector at $Y$ (2),[6]

$$\mathcal{A}(\mathrm{grad} f(Y) Y^\top) = 0, \qquad\qquad (11)$$

and since it is the orthogonal projection of $\nabla f(Y)$ to the tangent space, there exists $\mu \in \mathbb{R}^m$ such that

$$\mathrm{grad} f(Y) + 2\mathcal{A}^*(\mu)Y = \nabla f(Y) = 2CY, \qquad\qquad (12)$$

where $\mathcal{A}^* \colon \mathbb{R}^m \to \mathbb{S}^{n \times n}$ is the adjoint of $\mathcal{A}$. Indeed, considering symmetric matrices $A_1, \ldots, A_m$ such that $\mathcal{A}(X)_i = \langle A_i, X \rangle$, matrices $\mathcal{A}^*(\mu)Y = \mu_1 A_1 Y + \cdots + \mu_m A_m Y$ span the normal space to the manifold at $Y$. Right-multiply (12) with $Y^\top$ and apply $\mathcal{A}$ to obtain

$$\mathcal{A}\left(\mathcal{A}^*(\mu)YY^\top\right) = \mathcal{A}(CYY^\top). \qquad\qquad (13)$$

Under the assumption that the $A_i Y$'s are linearly independent, $\mu$ is the unique solution to this linear system—for KKT points, these are the Lagrange multipliers. Furthermore, contrary to classical KKT conditions, $\mu$ is defined for *all* feasible $Y$ (not only for KKT points) and can be found by solving (13).[7] This $\mu$ is a well-defined, differentiable function of $Y$.[8] Using this definition of $\mu$, let

$$S = S(Y) = S(YY^\top) = C - \mathcal{A}^*(\mu). \qquad\qquad (14)$$

First-order critical points then satisfy (using (12)):

$$\frac{1}{2}\mathrm{grad} f(Y) = SY = 0. \qquad\qquad (15)$$

We note in passing that $\mu(Y)$ is feasible for the dual of (SDP) exactly when $S(Y) \succeq 0$:

$$d^* = \max_{\mu \in \mathbb{R}^m} b^\top \mu \text{ subject to } C - \mathcal{A}^*(\mu) \succeq 0, \qquad\qquad \text{(DSDP)}$$

which illustrates the importance of $S$ as a dual certificate for (SDP).

Now let us turn to the Hessian of $f$. Equation (10) requires computation of the differential of $\mathrm{grad} f(Y)$, which is

$$\mathrm{D}\left(Y \mapsto \mathrm{grad} f(Y)\right)(Y)[\dot{Y}] = \mathrm{D}\left(Y \mapsto 2SY\right)(Y)[\dot{Y}] = 2S\dot{Y} + 2\dot{S}Y,$$

where $\dot{S} \triangleq \mathrm{D}S(Y)[\dot{Y}]$ is a symmetric matrix. Because of eq. (14), $\dot{S} = \mathcal{A}^*(\nu)$ for some $\nu \in \mathbb{R}^m$. Hence, for any tangent vector $\dot{Z} \in \mathrm{T}_Y \mathcal{M}$ (2), we have $\langle \dot{Z}, \dot{S}Y \rangle = \langle \dot{Z}Y^\top, \mathcal{A}^*(\nu) \rangle = \langle \mathcal{A}(\dot{Z}Y^\top), \nu \rangle = 0$: $\dot{S}Y$ is orthogonal to the tangent space at $Y$. Using (10), we find that

$$\frac{1}{2}\mathrm{Hess} f(Y)[\dot{Y}] = \mathrm{Proj}_Y S\dot{Y}. \qquad\qquad (16)$$

The second-order condition for $Y$ is that $\mathrm{Hess}f(Y)$ be positive semidefinite on $\mathrm{T}_Y\mathcal{M}$. Using that $\mathrm{Proj}_Y$ is a self-adjoint operator, it follows that this condition is equivalent to:

$$\forall \dot{Y} \in \mathrm{T}_Y\mathcal{M}, \quad \frac{1}{2}\langle \dot{Y}, \mathrm{Hess}f(Y)[\dot{Y}]\rangle = \langle \dot{Y}, S\dot{Y}\rangle \geq 0. \tag{17}$$

We now show that *rank-deficient* second-order critical points are globally optimal. We obtain this result as a corollary to a more informative statement about optimality gap at approximately second-order critical points (assuming exact rank deficiency). The lemmas also show how $S$ can be used to control the optimality gap at approximate critical points without requiring rank deficiency. This is valid for any $p$ and any $C$.

**Lemma 6.** *For any $Y$ on the manifold $\mathcal{M}$, if $\|\mathrm{grad}f(Y)\| \leq \varepsilon_g$ and $S(Y) \succeq -\frac{\varepsilon_H}{2}I_n$, then the optimality gap at $Y$ with respect to* (SDP) *is bounded as*

$$0 \leq 2(f(Y) - f^*) \leq \varepsilon_H R + \varepsilon_g\sqrt{R}, \tag{18}$$

*where $R = \max_{X\in\mathcal{C}}\mathrm{Tr}(X) < \infty$ measures the size of the compact set $\mathcal{C}$ (6). If $I_n \in \mathrm{im}(\mathcal{A}^*)$, the right hand side of (18) simplifies to $\varepsilon_H R$. This holds in particular if all $X \in \mathcal{C}$ have same trace and $\mathcal{C}$ has a relative interior point (Slater condition).*

*Proof.* By assumption on $S(Y)$ (eq. (14)),

$$\forall \tilde{X} \in \mathcal{C}, \quad -\frac{\varepsilon_H}{2}\mathrm{Tr}(\tilde{X}) \leq \langle S(Y), \tilde{X}\rangle = \langle C, \tilde{X}\rangle - \langle \mathcal{A}^*(\mu(Y)), \tilde{X}\rangle = \langle C, \tilde{X}\rangle - \langle \mu(Y), b\rangle.$$

This holds in particular for $\tilde{X}$ optimal for (SDP). Thus, we may set $\langle C, \tilde{X}\rangle = f^*$; and certainly, $\mathrm{Tr}(\tilde{X}) \leq R$. Furthermore,

$$\langle \mu(Y), b\rangle = \langle \mu(Y), \mathcal{A}(YY^\top)\rangle = \langle C - S(Y), YY^\top\rangle = f(Y) - \langle S(Y)Y, Y\rangle.$$

Combining the typeset equations and using $\mathrm{grad}f(Y) = 2S(Y)Y$, we find

$$0 \leq 2(f(Y) - f^*) \leq \varepsilon_H R + \langle \mathrm{grad}f(Y), Y\rangle. \tag{19}$$

In general, we do not assume $I_n \in \mathrm{im}(\mathcal{A}^*)$ and we get the result by Cauchy–Schwarz on (19) and $\|Y\| = \sqrt{\mathrm{Tr}(YY^\top)} \leq \sqrt{R}$:

$$0 \leq 2(f(Y) - f^*) \leq \varepsilon_H R + \varepsilon_g\sqrt{R}.$$

But if $I_n \in \mathrm{im}(\mathcal{A}^*)$, then we show that $Y$ is a normal vector at $Y$, so that it is orthogonal to $\mathrm{grad}f(Y)$. Formally: there exists $\nu \in \mathbb{R}^m$ such that $I_n = \mathcal{A}^*(\nu)$, and

$$\langle \mathrm{grad}f(Y), Y\rangle = \langle \mathrm{grad}f(Y)Y^\top, I_n\rangle = \langle \mathcal{A}(\mathrm{grad}f(Y)Y^\top), \nu\rangle = 0,$$

since $\mathrm{grad}f(Y) \in \mathrm{T}_Y\mathcal{M}$ (2). This indeed allows to simplify (19).

To conclude, we show that if $\mathcal{C}$ has a relative interior point $X'$ (that is, $\mathcal{A}(X') = b$ and $X' \succ 0$) and if $\mathrm{Tr}(X)$ is a constant for all $X$ in $\mathcal{C}$, then $I_n \in \mathrm{im}(\mathcal{A}^*)$. Indeed, $\mathbb{S}^{n\times n} = \mathrm{im}(\mathcal{A}^*) \oplus \ker\mathcal{A}$, so there exist $\nu \in \mathbb{R}^m$ and $M \in \ker\mathcal{A}$ such that $I_n = \mathcal{A}^*(\nu) + M$. Thus, for all $X$ in $\mathcal{C}$,

$$0 = \mathrm{Tr}(X - X') = \langle \mathcal{A}^*(\nu) + M, X - X'\rangle = \langle M, X - X'\rangle.$$

This implies that $M$ is orthogonal to all $X - X'$. These span $\ker\mathcal{A}$ since $X'$ is interior. Indeed, for any $H \in \ker\mathcal{A}$, since $X' \succ 0$, there exists $\varepsilon > 0$ such that $X \triangleq X' + \varepsilon H \succeq 0$ and $\mathcal{A}(X) = b$, so that $X \in \mathcal{C}$. Hence, $M \in \ker\mathcal{A}$ is orthogonal to $\ker\mathcal{A}$. Consequently, $M = 0$ and $I_n = \mathcal{A}^*(\nu)$. $\square$

**Lemma 7.** *If $Y \in \mathcal{M}$ is column rank deficient and $\mathrm{Hess}f(Y) \succeq -\varepsilon_H\,\mathrm{Id}$, then $S(Y) \succeq -\frac{\varepsilon_H}{2}I_n$.*

*Proof.* By assumption, there exists $z \in \mathbb{R}^p$, $\|z\| = 1$ such that $Yz = 0$. Thus, for any $x \in \mathbb{R}^n$, we can form $\dot{Y} = xz^\top$: it is a tangent vector since $Y\dot{Y}^\top = 0$ (2). Then, condition (17) combined with the assumption on $\mathrm{Hess}f(Y)$ tells us

$$-\varepsilon_H\|x\|^2 \leq \langle \dot{Y}, \mathrm{Hess}f(Y)[\dot{Y}]\rangle = 2\langle \dot{Y}, S\dot{Y}\rangle = 2\langle xz^\top zx^\top, S\rangle = 2x^\top Sx.$$

This holds for all $x \in \mathbb{R}^n$, hence $S \succeq -\frac{\varepsilon_H}{2}I_n$ as required. $\square$

**Corollary 8.** *If $Y \in \mathcal{M}_p$ is a column rank-deficient second-order critical point for* (P), *then it is optimal for* (P) *and $X = YY^\top$ is optimal for* (SDP). *In particular, for $p > n$, all second-order critical points are optimal.*

The first part of this corollary also appears as [Burer and Monteiro, 2003, Prop. 4], where the statement is made about local optima rather than second-order critical points.

At this point, we can already give a short proof of Theorem 4.

*Proof of Theorem 4.* Since $\tilde{Y}\tilde{Y}^\top = YY^\top$, $S(\tilde{Y}) = S(Y)$; in particular, $f(\tilde{Y}) = f(Y)$ and $\|\operatorname{grad} f(\tilde{Y})\| = \|\operatorname{grad} f(Y)\|$. Since $\tilde{Y}$ has deficient column rank, apply Lemmas 6 and 7. For $p > n$, there is no need to form $\tilde{Y}$ as $Y$ necessarily has deficient column rank. $\qquad\square$

Based on Corollary 8, to establish Theorem 2 it is sufficient to show that, for almost all $C$, all second-order critical points are rank deficient already for small $p$. We show that in fact this is true even for first-order critical points. The argument is by dimensionality counting on $\mathbb{S}^{n \times n}$: the set of all possible cost matrices $C$.

**Lemma 9.** *Under the assumptions of Theorem 2, for almost all $C$, all critical points of* (P) *are rank deficient.*

*Proof.* Let $Y$ be a critical point for (P). By the first-order condition $S(Y)Y = 0$ (15) and the definition of $S(Y) = C - \mathcal{A}^*(\mu(Y))$ (14), there exists $\mu \in \mathbb{R}^m$ such that

$$\operatorname{rank} Y \leq \operatorname{null}(C - \mathcal{A}^*(\mu)) \leq \max_{\nu \in \mathbb{R}^m} \operatorname{null}(C - \mathcal{A}^*(\nu)), \tag{20}$$

where null denotes the nullity (dimension of the kernel). This first step in the proof is inspired by [Wen and Yin, 2013, Thm. 3]. If the right hand side evaluates to $\ell$, then there exists $\nu$ such that $M = C - \mathcal{A}^*(\nu)$ and $\operatorname{null}(M) = \ell$. Writing $C = M + \mathcal{A}^*(\nu)$, we find that

$$C \in \mathcal{N}_\ell + \operatorname{im}(\mathcal{A}^*) \tag{21}$$

where the $+$ is a set-sum and $\mathcal{N}_\ell$ denotes the set of symmetric matrices of size $n$ with nullity $\ell$. This set has dimension

$$\dim \mathcal{N}_\ell = \frac{n(n+1)}{2} - \frac{\ell(\ell+1)}{2}, \tag{22}$$

whereas $\dim \operatorname{im}(\mathcal{A}^*) = \operatorname{rank}(\mathcal{A}^*) \leq m$. Assume the right hand side of (20) evaluates to $p$ or more. Then, a fortiori,

$$C \in \bigcup_{\ell=p,\dots,n} \mathcal{N}_\ell + \operatorname{im}(\mathcal{A}^*). \tag{23}$$

The set on the right hand side contains all "bad" $C$'s, that is, those for which (20) offers no information about the rank of $Y$. The dimension of that set is bounded as follows, using that the dimension of a finite union is at most the maximal dimension, and the dimension of a finite sum of sets is at most the sum of the set dimensions:

$$\dim \left( \bigcup_{\ell=p,\dots,n} \mathcal{N}_\ell + \operatorname{im}(\mathcal{A}^*) \right) \leq \dim \left( \mathcal{N}_p + \operatorname{im}(\mathcal{A}^*) \right) \leq \frac{n(n+1)}{2} - \frac{p(p+1)}{2} + m.$$

Since $C \in \mathbb{S}^{n \times n}$ lives in a space of dimension $\frac{n(n+1)}{2}$, almost no $C$ verifies (23) if

$$\frac{n(n+1)}{2} - \frac{p(p+1)}{2} + m < \frac{n(n+1)}{2}.$$

Hence, if $\frac{p(p+1)}{2} > m$, then, for almost all $C$, critical points verify $\operatorname{rank}(Y) < p$. $\qquad\square$

Theorem 2 follows as a corollary of Corollary 8 and Lemma 9.

# B Numerical experiments

As an example, we run five different solvers on (Max-Cut SDP) with a collection of graphs used in [Burer and Monteiro, 2003, 2005] known as the Gset.[9] The solvers are as follows, all run in Matlab. The first three are based on a low-rank factorization while the last two are interior point methods (IPM).

> Manopt runs the Riemannian Trust-Region method on (Max-Cut BM), via the Manopt toolbox [Boumal et al., 2014], with $p = \left\lceil \frac{\sqrt{8n+1}}{2} \right\rceil$ and random initialization. The number of inner iterations allowed to solve the trust-region subproblem is 500. The solver returns when $\frac{1}{2}\|\mathrm{grad}f(Y)\| = \|SY\| \leq 10^{-6}$. Code is in Matlab.

> Manopt+ runs the same algorithm as above, but with $p$ increasing from 2 to $\left\lceil \frac{\sqrt{8n+1}}{2} \right\rceil$ in 5 steps. The point $Y$ computed at a lower $p$ is appended with columns of i.i.d. random Gaussian variables with standard deviation $10^{-5}$ and mean 0, then rows are normalized to produce $Y_+$: the initial point for the next value of $p$. The randomization allows to escape near-saddle points (in practice). Code is in Matlab.

> SDPLR runs the original Burer–Monteiro algorithm implemented by its authors [Burer and Monteiro, 2003]. Code is in C interfaced through C-mex.

> HRVW runs an IPM whose implementation is tailored to (Max-Cut SDP), implemented by its authors [Helmberg et al., 1996]. Code is in Matlab.

> CVX runs SDPT3 [Toh et al., 1999] on (Max-Cut SDP) via CVX [CVX, 2012]. Code is in C interfaced through C-mex.

After the solvers return, we project their answers to the feasible set. Manopt and SDPLR return a matrix $Y$: it is sufficient to normalize each row to ensure $X = YY^\top$ is feasible for (Max-Cut SDP) (for Manopt, this step is not necessary). HRVW and CVX return a symmetric matrix $X$. We compute its Cholesky factorization $X = RR^\top$—if $X$ is not positive semidefinite, we first project using an eigenvalue decomposition. Then, each row of $R$ is normalized so that $X = RR^\top$ is feasible for (Max-Cut SDP). Computation time required for these projections is not included in the timings.

We report three metrics for each graph and each solver.

> Cut bound: a bound on the maximal cut value (lower is better). If $C$ is the adjacency matrix of the graph and $D$ is the degree matrix, then $L = D - C$ is the Laplacian and $\max_X \frac{1}{4} \langle L, X \rangle$ s.t. $\mathrm{diag}(X) = \mathbf{1}, X \succeq 0$ is a bound on the maximal cut. Using Lemma 6 applied to (Max-Cut SDP), a candidate optimizer $X$ yields a bound $\frac{1}{4} \langle L, X \rangle - \frac{n}{4}\lambda_{\min}(S)$.

> $\lambda_{\min}(S)$: by Lemma 6, this is a measure of optimality for $X$ (feasible), where $S = C - \mathrm{diag}(\mathrm{diag}(CX))$. It is nonpositive and must be as close to 0 as possible. We compute it using bisection and the Cholesky factorization to ensure accuracy.

> Time: computation time in seconds for the solver to run[10] (this excludes time taken to project the solution to the feasible set and to compute the reported metrics.)

Based on the results reported in Table 1, we make the following main observations: (i) the Manopt approach (optimization on manifolds, also advocated in [Journée et al., 2010]) consistently reaches high accuracy solutions, being often orders of magnitude more accurate than other methods, as judged from $\lambda_{\min}(S)$; (ii) incremental rank solvers (Manopt+ and SDPLR) are often the fastest solvers for large instances; and (iii) the tailored IPM HRVW is faster and typically more accurate than the IPM called by CVX (which is generic software). The latter point hints that one must be careful in dismissing IPMs based on experiments using generic software, although it remains clear from Table 1 that IPMs scale poorly compared to the low-rank factorization methods tested here. In particular, CVX runs into memory trouble for the larger problem instances reported.[11] To save time, we did not run CVX on the largest graphs.

# C  Numerical experiments: results

Table 1: Results of the experiments described in Section B.

| Graph | Metric | Manopt | Manopt+ | SDPLR | HRVW | CVX |
|---|---|---|---|---|---|---|
| Graph 1 | Cut bound | 12083.2 | 12083.2 | 12083.2 | 12083.2 | 12083.2 |
| 800 nodes | $\lambda_{\min}(S)$ | $-3 \cdot 10^{-11}$ | $-2 \cdot 10^{-11}$ | $-9 \cdot 10^{-6}$ | $-2 \cdot 10^{-5}$ | $-3 \cdot 10^{-6}$ |
| 19176 edges | Time [s] | 2.1 | 3.2 | 6.6 | 1.9 | 35.0 |
| Graph 2 | Cut bound | 12089.4 | 12089.4 | 12089.4 | 12089.4 | 12089.4 |
| 800 nodes | $\lambda_{\min}(S)$ | $-2 \cdot 10^{-10}$ | $-8 \cdot 10^{-12}$ | $-5 \cdot 10^{-6}$ | $-3 \cdot 10^{-5}$ | $-7 \cdot 10^{-7}$ |
| 19176 edges | Time [s] | 1.6 | 3.1 | 7.8 | 2.0 | 33.7 |
| Graph 3 | Cut bound | 12084.3 | 12084.3 | 12085.5 | 12084.3 | 12084.3 |
| 800 nodes | $\lambda_{\min}(S)$ | $-3 \cdot 10^{-11}$ | $-1 \cdot 10^{-11}$ | $-6 \cdot 10^{-3}$ | $-4 \cdot 10^{-5}$ | $-2 \cdot 10^{-6}$ |
| 19176 edges | Time [s] | 2.1 | 4.5 | 9.8 | 2.0 | 34.0 |
| Graph 4 | Cut bound | 12111.5 | 12111.5 | 12111.5 | 12111.5 | 12111.5 |
| 800 nodes | $\lambda_{\min}(S)$ | $-2 \cdot 10^{-11}$ | $-2 \cdot 10^{-10}$ | $-1 \cdot 10^{-5}$ | $-3 \cdot 10^{-5}$ | $-6 \cdot 10^{-6}$ |
| 19176 edges | Time [s] | 1.8 | 3.2 | 10.6 | 2.2 | 33.7 |
| Graph 5 | Cut bound | 12099.9 | 12099.9 | 12099.9 | 12099.9 | 12099.9 |
| 800 nodes | $\lambda_{\min}(S)$ | $-3 \cdot 10^{-12}$ | $-8 \cdot 10^{-12}$ | $-1 \cdot 10^{-5}$ | $-3 \cdot 10^{-5}$ | $-1 \cdot 10^{-6}$ |
| 19176 edges | Time [s] | 1.5 | 2.5 | 6.7 | 2.2 | 33.7 |
| Graph 6 | Cut bound | 2656.2 | 2656.2 | 2660.8 | 2656.2 | 2656.2 |
| 800 nodes | $\lambda_{\min}(S)$ | $-4 \cdot 10^{-12}$ | $-8 \cdot 10^{-12}$ | $-2 \cdot 10^{-2}$ | $-7 \cdot 10^{-6}$ | $-9 \cdot 10^{-6}$ |
| 19176 edges | Time [s] | 1.4 | 2.6 | 5.5 | 2.4 | 34.1 |
| Graph 7 | Cut bound | 2489.3 | 2489.3 | 2489.3 | 2489.3 | 2489.3 |
| 800 nodes | $\lambda_{\min}(S)$ | $-2 \cdot 10^{-11}$ | $-2 \cdot 10^{-11}$ | $-1 \cdot 10^{-5}$ | $-9 \cdot 10^{-6}$ | $-4 \cdot 10^{-7}$ |
| 19176 edges | Time [s] | 6.4 | 2.6 | 5.9 | 2.0 | 35.7 |
| Graph 8 | Cut bound | 2506.9 | 2506.9 | 2506.9 | 2506.9 | 2506.9 |
| 800 nodes | $\lambda_{\min}(S)$ | $-5 \cdot 10^{-12}$ | $-9 \cdot 10^{-12}$ | $-4 \cdot 10^{-5}$ | $-1 \cdot 10^{-5}$ | $-1 \cdot 10^{-6}$ |
| 19176 edges | Time [s] | 1.2 | 1.8 | 10.6 | 2.2 | 34.0 |
| Graph 9 | Cut bound | 2528.7 | 2528.7 | 2528.7 | 2528.7 | 2528.7 |
| 800 nodes | $\lambda_{\min}(S)$ | $-1 \cdot 10^{-9}$ | $-8 \cdot 10^{-12}$ | $-8 \cdot 10^{-6}$ | $-1 \cdot 10^{-5}$ | $-1 \cdot 10^{-6}$ |
| 19176 edges | Time [s] | 0.9 | 1.8 | 5.7 | 2.4 | 34.8 |
| Graph 10 | Cut bound | 2485.1 | 2485.1 | 2485.1 | 2485.1 | 2485.1 |
| 800 nodes | $\lambda_{\min}(S)$ | $-5 \cdot 10^{-11}$ | $-8 \cdot 10^{-12}$ | $-6 \cdot 10^{-6}$ | $-8 \cdot 10^{-6}$ | $-2 \cdot 10^{-6}$ |
| 19176 edges | Time [s] | 1.2 | 1.6 | 5.3 | 2.1 | 33.9 |
| Graph 11 | Cut bound | 629.2 | 629.2 | 629.2 | 629.2 | 629.2 |
| 800 nodes | $\lambda_{\min}(S)$ | $-3 \cdot 10^{-9}$ | $-7 \cdot 10^{-12}$ | $-5 \cdot 10^{-6}$ | $-1 \cdot 10^{-6}$ | $-4 \cdot 10^{-8}$ |
| 1600 edges | Time [s] | 13.6 | 13.6 | 3.9 | 2.0 | 31.5 |
| Graph 12 | Cut bound | 623.9 | 623.9 | 623.9 | 623.9 | 623.9 |
| 800 nodes | $\lambda_{\min}(S)$ | $-1 \cdot 10^{-10}$ | $-4 \cdot 10^{-12}$ | $-3 \cdot 10^{-6}$ | $-3 \cdot 10^{-6}$ | $-9 \cdot 10^{-8}$ |
| 1600 edges | Time [s] | 8.8 | 7.3 | 1.9 | 2.0 | 31.7 |
| Graph 13 | Cut bound | 647.1 | 647.1 | 647.1 | 647.1 | 647.1 |
| 800 nodes | $\lambda_{\min}(S)$ | $-1 \cdot 10^{-9}$ | $-2 \cdot 10^{-12}$ | $-2 \cdot 10^{-6}$ | $-2 \cdot 10^{-6}$ | $-1 \cdot 10^{-7}$ |
| 1600 edges | Time [s] | 6.9 | 6.7 | 1.3 | 2.2 | 31.4 |
| Graph 14 | Cut bound | 3191.6 | 3191.6 | 3191.6 | 3191.6 | 3191.6 |
| 800 nodes | $\lambda_{\min}(S)$ | $-1 \cdot 10^{-10}$ | $-3 \cdot 10^{-12}$ | $-3 \cdot 10^{-5}$ | $-3 \cdot 10^{-5}$ | $-1 \cdot 10^{-6}$ |
| 4694 edges | Time [s] | 1.5 | 5.3 | 4.4 | 2.5 | 34.1 |
| Graph 15 | Cut bound | 3171.6 | 3171.6 | 3171.6 | 3171.6 | 3171.6 |
| 800 nodes | $\lambda_{\min}(S)$ | $-1 \cdot 10^{-10}$ | $-5 \cdot 10^{-12}$ | $-6 \cdot 10^{-6}$ | $-5 \cdot 10^{-6}$ | $-3 \cdot 10^{-7}$ |
| 4661 edges | Time [s] | 3.4 | 6.5 | 5.4 | 3.2 | 34.6 |
| Graph 16 | Cut bound | 3175.0 | 3175.0 | 3175.1 | 3175.0 | 3175.0 |
| 800 nodes | $\lambda_{\min}(S)$ | $-9 \cdot 10^{-12}$ | $-2 \cdot 10^{-12}$ | $-6 \cdot 10^{-4}$ | $-1 \cdot 10^{-5}$ | $-6 \cdot 10^{-7}$ |
| 4672 edges | Time [s] | 6.6 | 6.2 | 3.8 | 3.1 | 34.8 |
| Graph 17 | Cut bound | 3171.3 | 3171.3 | 3171.5 | 3171.3 | 3171.3 |
| 800 nodes | $\lambda_{\min}(S)$ | $-5 \cdot 10^{-12}$ | $-2 \cdot 10^{-12}$ | $-1 \cdot 10^{-3}$ | $-1 \cdot 10^{-5}$ | $-1 \cdot 10^{-7}$ |
| 4667 edges | Time [s] | 6.1 | 6.3 | 3.5 | 2.9 | 34.5 |

| Graph | Metric | Manopt | Manopt+ | SDPLR | HRVW | CVX |
|---|---|---|---|---|---|---|
| Graph 18 | Cut bound | 1166.0 | 1166.0 | 1166.0 | 1166.0 | 1166.0 |
| 800 nodes | $\lambda_{\min}(S)$ | $-4 \cdot 10^{-12}$ | $-3 \cdot 10^{-12}$ | $-3 \cdot 10^{-6}$ | $-4 \cdot 10^{-6}$ | $-1 \cdot 10^{-6}$ |
| 4694 edges | Time [s] | 1.8 | 2.9 | 4.2 | 3.2 | 35.1 |
| Graph 19 | Cut bound | 1082.0 | 1082.0 | 1082.0 | 1082.0 | 1082.0 |
| 800 nodes | $\lambda_{\min}(S)$ | $-4 \cdot 10^{-10}$ | $-4 \cdot 10^{-12}$ | $-4 \cdot 10^{-6}$ | $-3 \cdot 10^{-6}$ | $-8 \cdot 10^{-7}$ |
| 4661 edges | Time [s] | 1.9 | 2.8 | 4.3 | 3.4 | 34.5 |
| Graph 20 | Cut bound | 1111.4 | 1111.4 | 1112.1 | 1111.4 | 1111.4 |
| 800 nodes | $\lambda_{\min}(S)$ | $-2 \cdot 10^{-12}$ | $-3 \cdot 10^{-12}$ | $-3 \cdot 10^{-3}$ | $-4 \cdot 10^{-6}$ | $-2 \cdot 10^{-6}$ |
| 4672 edges | Time [s] | 2.8 | 3.7 | 2.9 | 3.6 | 34.1 |
| Graph 21 | Cut bound | 1104.3 | 1104.3 | 1104.3 | 1104.3 | 1104.3 |
| 800 nodes | $\lambda_{\min}(S)$ | $-2 \cdot 10^{-11}$ | $-6 \cdot 10^{-12}$ | $-4 \cdot 10^{-6}$ | $-2 \cdot 10^{-6}$ | $-6 \cdot 10^{-6}$ |
| 4667 edges | Time [s] | 2.7 | 4.3 | 3.5 | 3.7 | 34.1 |
| Graph 22 | Cut bound | 14135.9 | 14135.9 | 14136.0 | 14135.9 | 14137.2 |
| 2000 nodes | $\lambda_{\min}(S)$ | $-8 \cdot 10^{-12}$ | $-8 \cdot 10^{-12}$ | $-3 \cdot 10^{-5}$ | $-3 \cdot 10^{-5}$ | $-2 \cdot 10^{-3}$ |
| 19990 edges | Time [s] | 5.5 | 4.9 | 22.5 | 25.7 | 177.7 |
| Graph 23 | Cut bound | 14142.1 | 14142.1 | 14142.1 | 14142.1 | 14143.5 |
| 2000 nodes | $\lambda_{\min}(S)$ | $-2 \cdot 10^{-11}$ | $-3 \cdot 10^{-11}$ | $-8 \cdot 10^{-6}$ | $-3 \cdot 10^{-5}$ | $-3 \cdot 10^{-3}$ |
| 19990 edges | Time [s] | 7.0 | 9.1 | 16.3 | 23.8 | 182.8 |
| Graph 24 | Cut bound | 14140.9 | 14140.9 | 14140.9 | 14140.9 | 14142.1 |
| 2000 nodes | $\lambda_{\min}(S)$ | $-1 \cdot 10^{-11}$ | $-7 \cdot 10^{-12}$ | $-1 \cdot 10^{-5}$ | $-2 \cdot 10^{-5}$ | $-2 \cdot 10^{-3}$ |
| 19990 edges | Time [s] | 4.5 | 5.7 | 24.3 | 24.8 | 173.3 |
| Graph 25 | Cut bound | 14144.2 | 14144.2 | 14148.8 | 14144.2 | 14145.8 |
| 2000 nodes | $\lambda_{\min}(S)$ | $-1 \cdot 10^{-9}$ | $-9 \cdot 10^{-12}$ | $-9 \cdot 10^{-3}$ | $-9 \cdot 10^{-6}$ | $-3 \cdot 10^{-3}$ |
| 19990 edges | Time [s] | 4.8 | 18.1 | 16.7 | 23.8 | 175.0 |
| Graph 26 | Cut bound | 14132.9 | 14132.9 | 14132.9 | 14132.9 | 14134.2 |
| 2000 nodes | $\lambda_{\min}(S)$ | $-7 \cdot 10^{-12}$ | $-1 \cdot 10^{-11}$ | $-4 \cdot 10^{-6}$ | $-2 \cdot 10^{-5}$ | $-3 \cdot 10^{-3}$ |
| 19990 edges | Time [s] | 6.8 | 6.5 | 14.4 | 23.1 | 177.6 |
| Graph 27 | Cut bound | 4141.7 | 4141.7 | 4145.0 | 4141.7 | 4143.1 |
| 2000 nodes | $\lambda_{\min}(S)$ | $-1 \cdot 10^{-11}$ | $-7 \cdot 10^{-12}$ | $-7 \cdot 10^{-3}$ | $-9 \cdot 10^{-6}$ | $-3 \cdot 10^{-3}$ |
| 19990 edges | Time [s] | 3.7 | 4.4 | 10.8 | 23.5 | 175.9 |
| Graph 28 | Cut bound | 4100.8 | 4100.8 | 4100.8 | 4100.8 | 4102.2 |
| 2000 nodes | $\lambda_{\min}(S)$ | $-2 \cdot 10^{-9}$ | $-6 \cdot 10^{-12}$ | $-3 \cdot 10^{-5}$ | $-7 \cdot 10^{-6}$ | $-3 \cdot 10^{-3}$ |
| 19990 edges | Time [s] | 3.0 | 8.0 | 19.6 | 26.5 | 176.8 |
| Graph 29 | Cut bound | 4208.9 | 4208.9 | 4208.9 | 4208.9 | 4210.0 |
| 2000 nodes | $\lambda_{\min}(S)$ | $-2 \cdot 10^{-11}$ | $-2 \cdot 10^{-11}$ | $-5 \cdot 10^{-6}$ | $-2 \cdot 10^{-6}$ | $-2 \cdot 10^{-3}$ |
| 19990 edges | Time [s] | 12.2 | 8.3 | 17.7 | 24.5 | 180.6 |
| Graph 30 | Cut bound | 4215.4 | 4215.4 | 4215.4 | 4215.4 | 4216.6 |
| 2000 nodes | $\lambda_{\min}(S)$ | $-7 \cdot 10^{-11}$ | $-6 \cdot 10^{-12}$ | $-5 \cdot 10^{-6}$ | $-6 \cdot 10^{-6}$ | $-2 \cdot 10^{-3}$ |
| 19990 edges | Time [s] | 19.8 | 10.5 | 11.6 | 25.2 | 176.7 |
| Graph 31 | Cut bound | 4116.7 | 4116.7 | 4119.1 | 4116.7 | 4118.0 |
| 2000 nodes | $\lambda_{\min}(S)$ | $-2 \cdot 10^{-11}$ | $-5 \cdot 10^{-12}$ | $-5 \cdot 10^{-3}$ | $-7 \cdot 10^{-6}$ | $-3 \cdot 10^{-3}$ |
| 19990 edges | Time [s] | 4.1 | 8.9 | 16.2 | 26.2 | 170.6 |
| Graph 32 | Cut bound | 1567.6 | 1567.6 | 1567.6 | 1567.6 | 1567.8 |
| 2000 nodes | $\lambda_{\min}(S)$ | $-2 \cdot 10^{-10}$ | $-8 \cdot 10^{-12}$ | $-1 \cdot 10^{-6}$ | $-1 \cdot 10^{-6}$ | $-3 \cdot 10^{-4}$ |
| 4000 edges | Time [s] | 45.6 | 25.4 | 13.9 | 21.7 | 142.6 |
| Graph 33 | Cut bound | 1544.3 | 1544.3 | 1544.3 | 1544.3 | 1544.4 |
| 2000 nodes | $\lambda_{\min}(S)$ | $-7 \cdot 10^{-10}$ | $-5 \cdot 10^{-12}$ | $-1 \cdot 10^{-6}$ | $-9 \cdot 10^{-7}$ | $-1 \cdot 10^{-4}$ |
| 4000 edges | Time [s] | 31.2 | 17.3 | 9.9 | 23.0 | 141.2 |
| Graph 34 | Cut bound | 1546.7 | 1546.7 | 1546.7 | 1546.7 | 1546.8 |
| 2000 nodes | $\lambda_{\min}(S)$ | $-1 \cdot 10^{-9}$ | $-5 \cdot 10^{-12}$ | $-2 \cdot 10^{-6}$ | $-1 \cdot 10^{-6}$ | $-2 \cdot 10^{-4}$ |
| 4000 edges | Time [s] | 31.3 | 22.0 | 7.7 | 23.6 | 143.9 |
| Graph 35 | Cut bound | 8014.7 | 8014.7 | 8014.7 | 8014.7 | 8015.3 |
| 2000 nodes | $\lambda_{\min}(S)$ | $-1 \cdot 10^{-9}$ | $-4 \cdot 10^{-11}$ | $-5 \cdot 10^{-6}$ | $-9 \cdot 10^{-6}$ | $-1 \cdot 10^{-3}$ |
| 11778 edges | Time [s] | 19.4 | 17.4 | 26.0 | 34.5 | 187.7 |

| Graph | Metric | Manopt | Manopt+ | SDPLR | HRVW | CVX |
|---|---|---|---|---|---|---|
| Graph 36 | Cut bound | 8006.0 | 8006.0 | 8006.0 | 8006.0 | 8006.6 |
| 2000 nodes | $\lambda_{\min}(S)$ | $-9 \cdot 10^{-10}$ | $-3 \cdot 10^{-11}$ | $-1 \cdot 10^{-5}$ | $-2 \cdot 10^{-5}$ | $-1 \cdot 10^{-3}$ |
| 11766 edges | Time [s] | 12.0 | 36.9 | 41.1 | 37.0 | 193.3 |
| Graph 37 | Cut bound | 8018.6 | 8018.6 | 8019.4 | 8018.6 | 8019.5 |
| 2000 nodes | $\lambda_{\min}(S)$ | $-2 \cdot 10^{-10}$ | $-1 \cdot 10^{-11}$ | $-1 \cdot 10^{-3}$ | $-1 \cdot 10^{-5}$ | $-2 \cdot 10^{-3}$ |
| 11785 edges | Time [s] | 11.2 | 15.4 | 38.4 | 35.2 | 191.1 |
| Graph 38 | Cut bound | 8015.0 | 8015.0 | 8015.0 | 8015.0 | 8015.5 |
| 2000 nodes | $\lambda_{\min}(S)$ | $-1 \cdot 10^{-10}$ | $-1 \cdot 10^{-11}$ | $-2 \cdot 10^{-5}$ | $-1 \cdot 10^{-5}$ | $-1 \cdot 10^{-3}$ |
| 11779 edges | Time [s] | 13.1 | 14.2 | 44.7 | 37.5 | 193.0 |
| Graph 39 | Cut bound | 2877.6 | 2877.6 | 2877.8 | 2877.6 | 2878.4 |
| 2000 nodes | $\lambda_{\min}(S)$ | $-4 \cdot 10^{-9}$ | $-7 \cdot 10^{-12}$ | $-3 \cdot 10^{-4}$ | $-4 \cdot 10^{-6}$ | $-2 \cdot 10^{-3}$ |
| 11778 edges | Time [s] | 16.9 | 12.2 | 31.9 | 39.3 | 195.8 |
| Graph 40 | Cut bound | 2864.8 | 2864.8 | 2866.2 | 2864.8 | 2865.6 |
| 2000 nodes | $\lambda_{\min}(S)$ | $-1 \cdot 10^{-11}$ | $-2 \cdot 10^{-11}$ | $-3 \cdot 10^{-3}$ | $-3 \cdot 10^{-6}$ | $-2 \cdot 10^{-3}$ |
| 11766 edges | Time [s] | 9.2 | 9.4 | 40.8 | 40.9 | 189.0 |
| Graph 41 | Cut bound | 2865.2 | 2865.2 | 2868.1 | 2865.2 | 2865.8 |
| 2000 nodes | $\lambda_{\min}(S)$ | $-4 \cdot 10^{-10}$ | $-1 \cdot 10^{-11}$ | $-6 \cdot 10^{-3}$ | $-4 \cdot 10^{-6}$ | $-1 \cdot 10^{-3}$ |
| 11785 edges | Time [s] | 5.3 | 8.6 | 30.8 | 40.9 | 189.8 |
| Graph 42 | Cut bound | 2946.3 | 2946.3 | 2948.3 | 2946.3 | 2947.0 |
| 2000 nodes | $\lambda_{\min}(S)$ | $-9 \cdot 10^{-12}$ | $-7 \cdot 10^{-12}$ | $-4 \cdot 10^{-3}$ | $-6 \cdot 10^{-6}$ | $-1 \cdot 10^{-3}$ |
| 11779 edges | Time [s] | 7.9 | 8.1 | 32.9 | 41.8 | 188.4 |
| Graph 43 | Cut bound | 7032.2 | 7032.2 | 7032.2 | 7032.2 | 7033.2 |
| 1000 nodes | $\lambda_{\min}(S)$ | $-3 \cdot 10^{-12}$ | $-4 \cdot 10^{-12}$ | $-6 \cdot 10^{-6}$ | $-2 \cdot 10^{-5}$ | $-4 \cdot 10^{-3}$ |
| 9990 edges | Time [s] | 1.9 | 2.3 | 3.6 | 3.8 | 36.4 |
| Graph 44 | Cut bound | 7027.9 | 7027.9 | 7029.2 | 7027.9 | 7029.4 |
| 1000 nodes | $\lambda_{\min}(S)$ | $-1 \cdot 10^{-8}$ | $-3 \cdot 10^{-12}$ | $-5 \cdot 10^{-3}$ | $-2 \cdot 10^{-5}$ | $-6 \cdot 10^{-3}$ |
| 9990 edges | Time [s] | 2.9 | 3.9 | 3.7 | 3.6 | 38.0 |
| Graph 45 | Cut bound | 7024.8 | 7024.8 | 7024.8 | 7024.8 | 7025.9 |
| 1000 nodes | $\lambda_{\min}(S)$ | $-1 \cdot 10^{-9}$ | $-5 \cdot 10^{-12}$ | $-2 \cdot 10^{-5}$ | $-8 \cdot 10^{-6}$ | $-5 \cdot 10^{-3}$ |
| 9990 edges | Time [s] | 1.3 | 6.1 | 4.9 | 3.5 | 37.4 |
| Graph 46 | Cut bound | 7029.9 | 7029.9 | 7029.9 | 7029.9 | 7030.8 |
| 1000 nodes | $\lambda_{\min}(S)$ | $-2 \cdot 10^{-10}$ | $-3 \cdot 10^{-12}$ | $-2 \cdot 10^{-5}$ | $-1 \cdot 10^{-5}$ | $-4 \cdot 10^{-3}$ |
| 9990 edges | Time [s] | 12.9 | 2.3 | 3.1 | 3.7 | 38.3 |
| Graph 47 | Cut bound | 7036.7 | 7036.7 | 7036.7 | 7036.7 | 7037.8 |
| 1000 nodes | $\lambda_{\min}(S)$ | $-8 \cdot 10^{-10}$ | $-9 \cdot 10^{-12}$ | $-1 \cdot 10^{-5}$ | $-1 \cdot 10^{-5}$ | $-5 \cdot 10^{-3}$ |
| 9990 edges | Time [s] | 10.4 | 4.1 | 8.2 | 3.8 | 39.2 |
| Graph 48 | Cut bound | 6000.0 | 6000.0 | 6000.0 | 6000.0 | 6000.0 |
| 3000 nodes | $\lambda_{\min}(S)$ | $4 \cdot 10^{-16}$ | $3 \cdot 10^{-16}$ | $-6 \cdot 10^{-10}$ | $-3 \cdot 10^{-6}$ | $5 \cdot 10^{-18}$ |
| 6000 edges | Time [s] | 2.8 | 4.3 | 3.5 | 47.7 | 307.3 |
| Graph 49 | Cut bound | 6000.0 | 6000.0 | 6000.0 | 6000.0 | 6000.0 |
| 3000 nodes | $\lambda_{\min}(S)$ | $4 \cdot 10^{-16}$ | $4 \cdot 10^{-16}$ | $-1 \cdot 10^{-9}$ | $-3 \cdot 10^{-6}$ | $-4 \cdot 10^{-16}$ |
| 6000 edges | Time [s] | 3.9 | 5.1 | 4.9 | 46.1 | 299.7 |
| Graph 50 | Cut bound | 5988.2 | 5988.2 | 5988.2 | 5988.2 | 5988.2 |
| 3000 nodes | $\lambda_{\min}(S)$ | $-2 \cdot 10^{-12}$ | $-1 \cdot 10^{-14}$ | $-1 \cdot 10^{-7}$ | $-3 \cdot 10^{-6}$ | $2 \cdot 10^{-16}$ |
| 6000 edges | Time [s] | 6.0 | 5.0 | 5.4 | 45.7 | 318.4 |
| Graph 51 | Cut bound | 4006.3 | 4006.3 | 4006.3 | 4006.3 | 4006.9 |
| 1000 nodes | $\lambda_{\min}(S)$ | $-2 \cdot 10^{-9}$ | $-4 \cdot 10^{-12}$ | $-8 \cdot 10^{-6}$ | $-1 \cdot 10^{-5}$ | $-3 \cdot 10^{-3}$ |
| 5909 edges | Time [s] | 5.8 | 7.8 | 10.7 | 5.4 | 41.4 |
| Graph 52 | Cut bound | 4009.6 | 4009.6 | 4010.0 | 4009.6 | 4010.2 |
| 1000 nodes | $\lambda_{\min}(S)$ | $-4 \cdot 10^{-12}$ | $-9 \cdot 10^{-12}$ | $-1 \cdot 10^{-3}$ | $-5 \cdot 10^{-6}$ | $-2 \cdot 10^{-3}$ |
| 5916 edges | Time [s] | 6.4 | 8.8 | 6.5 | 5.2 | 39.6 |
| Graph 53 | Cut bound | 4009.7 | 4009.7 | 4009.7 | 4009.7 | 4010.5 |
| 1000 nodes | $\lambda_{\min}(S)$ | $-1 \cdot 10^{-10}$ | $-1 \cdot 10^{-11}$ | $-6 \cdot 10^{-6}$ | $-1 \cdot 10^{-5}$ | $-3 \cdot 10^{-3}$ |
| 5914 edges | Time [s] | 4.2 | 8.5 | 8.3 | 5.0 | 39.1 |

| Graph | Metric | Manopt | Manopt+ | SDPLR | HRVW | CVX |
|---|---|---|---|---|---|---|
| Graph 54 | Cut bound | 4006.2 | 4006.2 | 4006.2 | 4006.2 | 4006.9 |
| 1000 nodes | $\lambda_{\min}(S)$ | $-2 \cdot 10^{-10}$ | $-3 \cdot 10^{-12}$ | $-3 \cdot 10^{-5}$ | $-5 \cdot 10^{-6}$ | $-3 \cdot 10^{-3}$ |
| 5916 edges | Time [s] | 2.9 | 6.6 | 6.1 | 4.8 | 39.1 |
| Graph 55 | Cut bound | 11039.5 | 11039.5 | 11039.5 | 11039.5 | 11039.7 |
| 5000 nodes | $\lambda_{\min}(S)$ | $-2 \cdot 10^{-12}$ | $-3 \cdot 10^{-12}$ | $-5 \cdot 10^{-6}$ | $-6 \cdot 10^{-6}$ | $-2 \cdot 10^{-4}$ |
| 12498 edges | Time [s] | 26.6 | 20.6 | 22.2 | 411.4 | 1588.0 |
| Graph 56 | Cut bound | 4760.0 | 4760.0 | 4760.0 | 4760.0 | 4760.3 |
| 5000 nodes | $\lambda_{\min}(S)$ | $-7 \cdot 10^{-12}$ | $-2 \cdot 10^{-12}$ | $-1 \cdot 10^{-5}$ | $-2 \cdot 10^{-6}$ | $-3 \cdot 10^{-4}$ |
| 12498 edges | Time [s] | 20.1 | 16.3 | 32.9 | 475.9 | 1550.1 |
| Graph 57 | Cut bound | 3885.5 | 3885.5 | 3885.5 | 3885.5 | 3885.7 |
| 5000 nodes | $\lambda_{\min}(S)$ | $-1 \cdot 10^{-9}$ | $-8 \cdot 10^{-12}$ | $-2 \cdot 10^{-6}$ | $-2 \cdot 10^{-6}$ | $-1 \cdot 10^{-4}$ |
| 10000 edges | Time [s] | 218.0 | 78.8 | 38.3 | 269.8 | 1012.4 |
| Graph 58 | Cut bound | 20136.2 | 20136.2 | 20138.1 | 20136.2 | 20136.7 |
| 5000 nodes | $\lambda_{\min}(S)$ | $-3 \cdot 10^{-9}$ | $-5 \cdot 10^{-11}$ | $-2 \cdot 10^{-3}$ | $-7 \cdot 10^{-6}$ | $-4 \cdot 10^{-4}$ |
| 29570 edges | Time [s] | 55.4 | 44.0 | 321.5 | 497.9 | 1865.7 |
| Graph 59 | Cut bound | 7312.3 | 7312.3 | 7315.0 | 7312.3 | 7313.0 |
| 5000 nodes | $\lambda_{\min}(S)$ | $-7 \cdot 10^{-12}$ | $-3 \cdot 10^{-11}$ | $-2 \cdot 10^{-3}$ | $-4 \cdot 10^{-6}$ | $-5 \cdot 10^{-4}$ |
| 29570 edges | Time [s] | 51.3 | 35.6 | 353.1 | 511.3 | 1869.0 |
| Graph 60 | Cut bound | 15222.3 | 15222.3 | 15222.3 | 15222.3 | 15222.6 |
| 7000 nodes | $\lambda_{\min}(S)$ | $-3 \cdot 10^{-11}$ | $-4 \cdot 10^{-12}$ | $-2 \cdot 10^{-5}$ | $-2 \cdot 10^{-6}$ | $-2 \cdot 10^{-4}$ |
| 17148 edges | Time [s] | 58.6 | 30.9 | 63.6 | 1326.9 | 3581.9 |
| Graph 61 | Cut bound | 6828.1 | 6828.1 | 6828.2 | 6828.1 | 6828.4 |
| 7000 nodes | $\lambda_{\min}(S)$ | $-2 \cdot 10^{-11}$ | $-4 \cdot 10^{-12}$ | $-7 \cdot 10^{-5}$ | $-2 \cdot 10^{-6}$ | $-2 \cdot 10^{-4}$ |
| 17148 edges | Time [s] | 113.4 | 40.2 | 55.8 | 1263.3 | 3795.6 |
| Graph 62 | Cut bound | 5430.9 | 5430.9 | 5430.9 | 5430.9 | 5431.1 |
| 7000 nodes | $\lambda_{\min}(S)$ | $-1 \cdot 10^{-9}$ | $-6 \cdot 10^{-11}$ | $-9 \cdot 10^{-7}$ | $-2 \cdot 10^{-6}$ | $-1 \cdot 10^{-4}$ |
| 14000 edges | Time [s] | 813.8 | 242.8 | 110.8 | 862.4 | 2124.3 |
| Graph 63 | Cut bound | 28244.4 | 28244.4 | 28245.9 | 28244.4 | 28245.0 |
| 7000 nodes | $\lambda_{\min}(S)$ | $-7 \cdot 10^{-9}$ | $-8 \cdot 10^{-9}$ | $-8 \cdot 10^{-4}$ | $-9 \cdot 10^{-6}$ | $-3 \cdot 10^{-4}$ |
| 41459 edges | Time [s] | 238.9 | 97.6 | 663.0 | 1454.7 | 4583.9 |
| Graph 64 | Cut bound | 10465.9 | 10465.9 | 10466.6 | 10465.9 | 10466.6 |
| 7000 nodes | $\lambda_{\min}(S)$ | $-3 \cdot 10^{-9}$ | $-2 \cdot 10^{-11}$ | $-4 \cdot 10^{-4}$ | $-5 \cdot 10^{-6}$ | $-4 \cdot 10^{-4}$ |
| 41459 edges | Time [s] | 140.4 | 109.5 | 1014.8 | 1609.4 | 4439.8 |
| Graph 65 | Cut bound | 6205.5 | 6205.5 | 6205.5 | 6205.5 | 6205.7 |
| 8000 nodes | $\lambda_{\min}(S)$ | $-1 \cdot 10^{-9}$ | $-1 \cdot 10^{-11}$ | $-6 \cdot 10^{-7}$ | $-1 \cdot 10^{-6}$ | $-1 \cdot 10^{-4}$ |
| 16000 edges | Time [s] | 567.2 | 168.5 | 154.4 | 1075.2 | 2861.5 |
| Graph 66 | Cut bound | 7077.2 | 7077.2 | 7077.2 | 7077.2 | 7077.4 |
| 9000 nodes | $\lambda_{\min}(S)$ | $-2 \cdot 10^{-9}$ | $-5 \cdot 10^{-11}$ | $-2 \cdot 10^{-7}$ | $-9 \cdot 10^{-7}$ | $-6 \cdot 10^{-5}$ |
| 18000 edges | Time [s] | 762.6 | 215.3 | 218.1 | 1525.7 | 3915.7 |
| Graph 67 | Cut bound | 7744.4 | 7744.4 | 7744.4 | 7744.4 | - |
| 10000 nodes | $\lambda_{\min}(S)$ | $-1 \cdot 10^{-9}$ | $-3 \cdot 10^{-11}$ | $-3 \cdot 10^{-7}$ | $-1 \cdot 10^{-6}$ | - |
| 20000 edges | Time [s] | 816.4 | 339.0 | 267.3 | 2005.4 | - |
| Graph 70 | Cut bound | 9861.5 | 9861.5 | 9861.5 | 9861.5 | - |
| 10000 nodes | $\lambda_{\min}(S)$ | $-2 \cdot 10^{-10}$ | $-6 \cdot 10^{-13}$ | $-2 \cdot 10^{-6}$ | $-2 \cdot 10^{-6}$ | - |
| 9999 edges | Time [s] | 143.3 | 82.9 | 102.2 | 3167.3 | - |
| Graph 72 | Cut bound | 7808.5 | 7808.5 | 7808.5 | 7808.5 | - |
| 10000 nodes | $\lambda_{\min}(S)$ | $-6 \cdot 10^{-10}$ | $-8 \cdot 10^{-12}$ | $-8 \cdot 10^{-7}$ | $-1 \cdot 10^{-6}$ | - |
| 20000 edges | Time [s] | 720.8 | 262.6 | 199.0 | 1902.7 | - |
| Graph 77 | Cut bound | 11045.7 | 11045.7 | 11045.7 | 11045.7 | - |
| 14000 nodes | $\lambda_{\min}(S)$ | $-8 \cdot 10^{-10}$ | $-4 \cdot 10^{-11}$ | $-7 \cdot 10^{-7}$ | $-1 \cdot 10^{-6}$ | - |
| 28000 edges | Time [s] | 1578.5 | 513.0 | 515.1 | 5249.1 | - |
| Graph 81 | Cut bound | 15656.2 | 15656.2 | 15656.2 | 15656.2 | - |
| 20000 nodes | $\lambda_{\min}(S)$ | $-5 \cdot 10^{-10}$ | $-6 \cdot 10^{-11}$ | $-1 \cdot 10^{-6}$ | $-3 \cdot 10^{-6}$ | - |
| 40000 edges | Time [s] | 4152.8 | 1539.7 | 1035.6 | 16576.6 | - |

## D   Regularity assumption

Originally, Theorems 2 and 4 had the assumption that the search space of the factorized problem,

$$\mathcal{M} = \{Y \in \mathbb{R}^{n \times p} : \mathcal{A}(YY^\top) = b\},$$

is a manifold. From this assumption, we stated incorrectly that the tangent space at $Y$ of $\mathcal{M}$, denoted by $\mathrm{T}_Y \mathcal{M}$, is given by (2):

$$\mathrm{T}_Y \mathcal{M} = \{\dot{Y} \in \mathbb{R}^{n \times p} : \mathcal{A}(\dot{Y}Y^\top + Y\dot{Y}^\top) = 0\}.$$

This identity is used in a number of places of the proofs. In general, $\mathcal{M}$ being an embedded submanifold of $\mathbb{R}^{n \times p}$ only implies the left hand side is included in the right hand side.[12] Below, we give an example where $\mathcal{M}$ is a manifold yet the two sets are not equal.

In order to restore equality, we strengthened the assumption, requiring constraint qualifications to hold at all feasible points (see (7)):

$$\forall Y \in \mathcal{M}, \quad A_1 Y, \dots, A_m Y \text{ are linearly independent in } \mathbb{R}^{n \times p},$$

where $A_i$, $i = 1, \dots, m$, are the symmetric constraint matrices such that $\mathcal{A}(X)_i = \langle A_i, X \rangle$. This ensures the map $\Phi(Y) = \mathcal{A}(YY^\top) - b$ is full rank on $\mathcal{M} = \Phi^{-1}(0)$, from which it follows by a standard result in differential geometry (see for example [Lee, 2012, Cor. 5.14]) that $\mathcal{M}$ is a smooth embedded submanifold of $\mathbb{R}^{n \times p}$ of dimension $np - m$. Then, the left hand side of (2) has dimension $np - m$, and it is included in the right hand side, which itself is a linear space of dimension $np - m$, so that they are equal.

We now describe an SDP such that $\mathcal{M}$ is indeed a manifold, yet (2) does not hold. Consider $n = 2, m = 2, b = (1,1)^\top$ and

$$A_1 = \begin{pmatrix} 1 & 0 \\ 0 & 1 \end{pmatrix}, \qquad\qquad A_2 = \begin{pmatrix} 1 & 0 \\ 0 & \frac{1}{4} \end{pmatrix}.$$

The search space of the SDP,

$$\mathcal{C} = \{X = X^\top \in \mathbb{R}^{n \times n} : \mathcal{A}(X) = b, X \succeq 0\} = \left\{ \begin{pmatrix} 1 & 0 \\ 0 & 0 \end{pmatrix} \right\},$$

is degenerate but it is compact. Furthermore, the set $\mathcal{M}$ is a smooth manifold for $p = 1$:

$$\mathcal{M}_{p=1} = \left\{ Y = \begin{pmatrix} y_1 \\ y_2 \end{pmatrix} \in \mathbb{R}^2 : y_1^2 + y_2^2 = 1 \text{ and } y_1^2 + \frac{1}{4} y_2^2 = 1 \right\} = \{(1,0)^\top, (-1,0)^\top\}.$$

The dimension of the manifold is 0, so that $\mathrm{T}_Y \mathcal{M} = \{0\}$ for all $Y \in \mathcal{M}$. Consider now the right hand side of (2),

$$K_Y = \{\dot{Y} \in \mathbb{R}^{n \times p} : \mathcal{A}(\dot{Y}Y^\top + Y\dot{Y}^\top) = 0\} = \{\dot{Y} \in \mathbb{R}^{n \times p} : \langle A_1 Y, \dot{Y} \rangle = \langle A_2 Y, \dot{Y} \rangle = 0\}.$$

These are the vectors orthogonal to $A_1 Y, A_2 Y$. For $Y = (\pm 1, 0)^\top$, we get $A_1 Y = A_2 Y = (\pm 1, 0)^\top$: they are colinear, so $K_Y$ has dimension 1 at all $Y \in \mathcal{M}$: $\mathrm{T}_Y \mathcal{M} \neq K_Y$.

Similarly, at $p = 2$, the set $\mathcal{M}$ becomes a circle embedded in $\mathbb{R}^4$:

$$\mathcal{M}_{p=2} = \left\{ Y = \begin{pmatrix} y_1 & y_2 \\ y_3 & y_4 \end{pmatrix} \in \mathbb{R}^{2 \times 2} : y_1^2 + y_2^2 + y_3^2 + y_4^2 = 1 \text{ and } y_1^2 + y_2^2 + \frac{1}{4}(y_3^2 + y_4^2) = 1 \right\}$$

$$= \left\{ Y = \begin{pmatrix} y_1 & y_2 \\ y_3 & y_4 \end{pmatrix} \in \mathbb{R}^{2 \times 2} : y_1^2 + y_2^2 = 1 \text{ and } y_3 = y_4 = 0 \right\}.$$

This manifold has dimension 1 (and so do all its tangent spaces). Yet, $K_Y$ has dimension 3 for all $Y \in \mathcal{M}$. Indeed, we can parameterize $\mathcal{M}_{p=2}$ as the matrices

$$\begin{pmatrix} \cos\theta & \sin\theta \\ 0 & 0 \end{pmatrix}$$

for all $\theta \in \mathbb{R}$. It is easy to verify that $A_1 Y = A_2 Y \neq 0$ for all $Y \in \mathcal{M}_{p=2}$, so that the codimension of $K_Y$ is 1, here too in disagreement with $\mathrm{T}_Y \mathcal{M}$. Notice also that in this example we have $\frac{p(p+1)}{2} > m$.