[Reviews · NeurIPS 2016]

Reviewer 1

Summary

Authors study the conditions, under which critical points of semidefinite programs coincide with optima. Then, they suggest an algorithm which obtains a critical point, test the conditions, and tries to escape the critical point, if needed. The algorithm is rather fancifully called the "Riemannian staircase". Although this is an important subject, it turns out that neither the algorithm nor any of the theorems are novel, really.

Qualitative Assessment

NIPS is not a good venue for a survey of recent work. I would suggest you submit to SIAM Review or write a book.

Confidence in this Review

3-Expert (read the paper in detail, know the area, quite certain of my opinion)


Reviewer 2

Summary

This paper builds upon Burer-Monteiro and Journee et al. to show that every critical point is rank deficient when p*(p+1)/2 >m. Furthermore, every second-order critical point that is rank deficient is the global optimum. By combining these two results, they conclude that every local minimum is global. The improvement over burer-monteiro is small, since the previous papers showed a similar result for p*(p+1)/2 >m. However, this paper establishes more quantitative guarantees like f(Y) -f* < sqrt(R) ||grad f(Y)||- R \lambda_min (Hess f(Y)). This kind of result leads to immediate algorithmic conclusions by combining with recent results on (stochastic) gradient and trust region.

Qualitative Assessment

The actual improvement over burer-monteiro is small, and also p*(p+1)/2 is a very pessimistic estimate of the true rank in practice. However I found the theoretical results to be interesting, and the method of proof to be different from burer-monteiro. I like the quantitative bounds provided in theorem 4 that lead to algorithmic conclusions.

Confidence in this Review

2-Confident (read it all; understood it all reasonably well)


Reviewer 3

Summary

In this paper, the authors consider a class of SDP and show that the low-rank Burer–Monteiro formulation of SDP’s in that class almost never has any spurious local optima.

Qualitative Assessment

This paper is well-written and properly organized. The authors are able to provide high probability results to characterize relations between a second-order critical point of a nonconvex problem (QCQP) and the optimal solution for SDP. Also, the properties of algorithm RTR which can converge to second-order critical points are further analyzed. In addition, the authors discuss in details the assumptions of the main theorem and justify the properness of these assumptions.

Confidence in this Review

2-Confident (read it all; understood it all reasonably well)


Reviewer 4

Summary

This paper provides a significant theory for solving a broad class of SDPs. It considers the recent Burer-Monteiro approach, which reduces SDPs to non-convex formulations with much smaller number of variables. The authors proved that in almost all cases any local minima of the non-convex programs are the global optimum of the original SDPs. This result is a significant theoretical support to the recent empirical success of the Burer-Monteiro approach. In addition, the authors show that the Riemannian trust-region method provides fast convergence to the local optima, which are the global optima of the original SDPs.

Qualitative Assessment

This paper will be a seminal result that will support theoretically the future Burer-Monteiro based algorithms.

Confidence in this Review

2-Confident (read it all; understood it all reasonably well)


Reviewer 5

Summary

This paper considers a specific class of semi-definite programs (minimize linear objective subject to linear equally and positive semi-definiteness constraints). This convex optimization problem is replaced by a non convex one where the optimization variable is replaced by the outer product of a matrix and its transpose. The paper shows that under certain conditions (size of the factorization large enough. compact and smooth search space), all second-order critical points are globally optimal. More specifically, the paper is very closely related to prior work by Burer and Monteiro. I think [1] proposed the change of variables X=YY^T, proposed also an BFGS augmented Lagrangian algorithm for solving the nonconvex problem, and showed experimentally that it reliably gives a global minimum if number of columns of Y is large enough. Then, [2] showed that the change of variables does not introduce extraneous local minima, that rank deficient local minima are global minima, and that a minor modification of BFGS is such that if a local minimum is obtained at each stage of the algorithm, then any accumulation point is a rank deficient local minima, and hence a global minima. So the main open problem with Burer and Monteiro appears to be that guaranteeing that the algorithm returns a local minima at each iteration. The core contribution of this paper is to address this issue. This is done by making two additional assumptions: (1) the set {X: A(X) = b} is compact, and (2) the set {Y: A(YY^T) = b} is a smooth manifold. These two assumptions allow the use of a Riemannian trust region method, which is guaranteed to converge to a second order critical point (new result from Boumal, P.-A. Absil, and C. Cartis in preparation which I did not have access to). As a consequence, it is not necessary to check that one is at a local minima at each step of the algorithm: one has guaranteed convergence to a second order critical point (as per new results from Bournal). Then, this paper shows that second-order critical points are globally optimal. So the core contribution is to not only show that rank-deficient local minima are global, but to also provide an algorithm that can find them. [1] S. Burer and R.D.C. Monteiro. A nonlinear programming algorithm for solving semidefinite programs via low-rank factorization. Mathematical Programming (Series B), 95:329–357, 2003. [2] S. Burer and R.D.C. Monteiro. Local minima and convergence in low-rank semidefinite programming. Mathematical Programming, 103(3):427–444, 2005.

Qualitative Assessment

I believe that (modulo some notational issues described below) the paper is well written, interesting, and worth knowing about. I found it interesting that the paper treats the constraint set as a manifold and uses Riemannian optimization techniques not just as an algorithm (Riemannian trust region) but also to establish conditions for optimality. Also, I liked that the paper provides specific bounds on how far the value of the objective is from the optimal value, and the fact that the framework is applicable to important applications, e.g., Max-cut. On the other hand, a key limitation of the paper is that it addresses a restricted set of optimization problems (sub-class of SDPs, whose constraint in factorized form is a compact and smooth manifold). In addition, the paper is less novel than it appears given recent results that are not discussed. In particular, [1] considers the potentially more general case where the matrix to be factorized is not symmetric, and so the factors are not constrained to be equal, the objective function is not constrained to be quadratic, [2] extends [1] to account for conic constraint and gauge functions, and [3] further extends the results to non-differentiable objectives and uses simpler alternating proximal gradient methods for minimizing the objective. While the results in this paper need need not be a corollary of [1,2,3], especially because [1,2,3] use some form of regularization, I believe a discussion of the differences would help further appreciate the novelty of the present paper since [1,2,3] also have results like “if the size of the factorization is large enough, local minima are global”. [1] F. Bach, J. Mairal, and J. Ponce, “Convex sparse matrix factorizations,” arXiv:0812.1869v1, 2008. [2] F. Bach, “Convex relaxations of structured matrix factorizations,” arXiv:1309.3117v1, 2013. [3] B. Haeffele, E. Young, and R. Vidal, “Structured low-rank matrix factorization: Optimality, algorithm, and applications to image processing,” in International Conference on Machine Learning, 2014. Other comments are that in the Proof of Theorem 4, S(Y) seems to be undefined. Then, in footnote 1, lambda_min(S) is mentioned, so S seems to be a matrix rather than a function. Overall, notation is undefined and sloppy at times. Minor comments: P2L33: The acronym L-BFGS has not been introduced. P2L70: The statement "Since (SDP) is a relaxation of (QCQP) (up to parameterization)” is not clear. In what sense is a relaxation and what parameterization are you referring to? P3L79: The statement "cost matrix admissible for our theory” is not unclear until one reads Theorem 1. Such terminology should be introduced in line 70.

Confidence in this Review

2-Confident (read it all; understood it all reasonably well)